# Spreading depression as an innate antiseizure mechanism

Isra Tamim [1,2], David Y. Chung [1,3], Andreia Lopes de Morais [1], Inge C. M. Loonen [1], Tao Qin[1], Amrit Misra[3], Frieder Schlunk[2], Matthias Endres [2], Steven J. Schiff [4] & Cenk Ayata [1,3✉]

Spreading depression (SD) is an intense and prolonged depolarization in the central nervous systems from insect to man. It is implicated in neurological disorders such as migraine and brain injury. Here, using an in vivo mouse model of focal neocortical seizures, we show that SD may be a fundamental defense against seizures. Seizures induced by topical 4-amino-pyridine, penicillin or bicuculline, or systemic kainic acid, culminated in SDs at a variable rate. Greater seizure power and area of recruitment predicted SD. Once triggered, SD immediately suppressed the seizure. Optogenetic or KCl-induced SDs had similar antiseizure effect sustained for more than 30 min. Conversely, pharmacologically inhibiting SD occurrence during a focal seizure facilitated seizure generalization. Altogether, our data indicate that seizures trigger SD, which then terminates the seizure and prevents its generalization.

[1] Neurovascular Research Unit, Department of Radiology, Massachusetts General Hospital, Harvard Medical School, Boston, MA, USA.
[2] Charité–Universitätsmedizin Berlin, Klinik und Hochschulambulanz für Neurologie und Centrum für Schlaganfallforschung Berlin (CSB), Berlin, Germany.
[3] Department of Neurology, Massachusetts General Hospital, Harvard Medical School, Boston, MA, USA. [4] Center for Neural Engineering, Departments of Engineering Science and Mechanics and Physics, The Pennsylvania State University, State College, PA, USA. ✉email: CAYATA@mgh.harvard.edu

Spreading depression (also known as spreading depolarization, SD) is an intense but self-limited neuronal and glial depolarization wave that slowly propagates (millimeters/minute) in the gray matter by way of chemical contiguity[1]. Near-total loss of transmembrane ion gradients during SD precludes action potential generation and silences synaptic transmission for more than a minute. These properties prompted Leão to name the phenomenon "spreading depression" more than 75 years ago[2].

All known triggers for SD are strong and sustained depolarizing events (e.g., ischemia, trauma) elevating extracellular $[K^+]$ above a critical threshold of 12 mM. Yet SD is also the physiological basis of migraine aura[3], and how and why such an intense depolarization event is initiated in ostensibly normal brains of migraine sufferers has perplexed scientists for decades. In fact, SD has been observed in the central nervous systems of all species studied to date from insect to man. It appears to be a fundamental intrinsic property of neurons and neuronal ensembles that somehow affords a survival advantage. The nature of the survival advantage (i.e., the selection pressure) and the significance of SD for normal brain function, however, have been a mystery.

Migraine aura and epilepsy are comorbid paroxysmal disorders, and hyperexcitability, also implicated in migraine, is one mechanism that can incite SD. For example, mutations in $Ca_V2.1$ voltage-gated calcium channels in familial hemiplegic migraine[4] and pharmacological disinhibition by $GABA_A$ receptor blockers[5] facilitate and trigger SD. Both are also known to predispose to seizures. Moreover, seizures and SD share common triggers[1], and seizures can culminate in SD both in experimental animals[6,7] and in humans[8]. Indeed, a mixed dynamic of seizure and SD is predicted by an extended Hodgkin–Huxley model incorporating cell volume changes and oxygen availability, suggesting that seizures and SD represent a dynamic continuum of neuronal membranes[9,10].

These observations raise the intriguing hypothesis that highly focal seizure activity triggers SD, and SD, by its electrical silencing effect, terminates the seizure. Here, we tested this hypothesis by undertaking a comprehensive examination of the reciprocal interactions between highly focal cortical seizures and SD. Our data provide the first direct proof-of-concept for an antiseizure role for SD and implicate highly focal seizures as potential triggers for migraine aura.

## Results
**Focal cortical seizures induced by topical application of epileptogenic agents.** In animals with topical vehicle application (0.9% NaCl), ECoG morphology and power density remained normal and stable for the entire duration of recordings (Fig. 1; setup as in Supplementary Fig. 1B, a). All three epileptogenic agents induced seizures within 10 min after topical application. Seizure morphology, duration, and the increase in ECoG power density within each frequency band differed among the three agents, recorded for a maximum of 4 h (Fig. 1, also see Supplementary Fig. 2 for ECoG at expanded time scale). Potassium channel blocker 4AP (100 mM) induced synchronized monomorphic spiking that gradually evolved into a periodic burst pattern. The electrographic pattern consisted of large amplitude (~400% of baseline) rhythmic discharges that started at ~1 Hz and evolved to ~4 Hz before attenuation. The cycle repeated itself over a ~30-min period. During bursting, the largest peak power density increase was within the δ frequency range (0–4 Hz, 7.3-fold) followed by θ (4–8 Hz, 3.2-fold) and α (8–12 Hz, 3.4-fold) bands. Seizures typically lasted well over 2 h with gradually decreasing synchronization and burst duration. In contrast, GABA-A receptor antagonist PG (188434 IU/ml) induced near-continuous high-amplitude periodic spikes that changed polarity

in a subset of experiments after $41 \pm 13$ min. The electrographic pattern started as ~0.5 Hz spike-wave activity with an amplitude ~250% over baseline and evolved in amplitude to ~600% over baseline with continuous ~0.5 Hz activity. This pattern developed over a ~70-min period. The largest peak power density increase with PG was within the α (13.7-fold) and θ (12.4-fold) bands followed by δ (3.6-fold) band; power increase in the θ band lasted for the duration of the recordings (4 h), while α band power increase resolved within 90 min. The second GABA-A receptor antagonist BIC (5 mM) induced seizures that were similar to PG in morphology and peak power density increase (19.9-fold in α, 13.3-fold in θ, 3.4-fold in δ), but spikes typically remained negative and lasted less than an hour (note time scale on Fig. 1). The electrographic pattern started as ~0.2 Hz spike-wave activity with an amplitude of ~120% of baseline and evolved into ~2 Hz spike-wave activity with an amplitude of ~250%. This pattern developed over a ~40-min period. The epileptogenic effect was concentration-dependent when tested for 4AP and BIC.

**Focal cortical seizures trigger spreading depression.** Neocortical seizures culminated in one or more SDs associated with large slow (DC) potential shifts in the seizure focus (Fig. 2; setup as in Supplementary Fig. 1B, a). These always propagated to the remote ipsilateral recording site, coupled to characteristic CBV transients on IOS imaging (Supplementary Movie 1). The highest proportion of animals that developed SD was during 4AP-induced seizures (69%, $n = 32$ mice), followed by PG (33%, $n = 9$ mice) and BIC (9%, $n = 11$ mice; $p = 0.002$, $\chi^2$; Fig. 3A). Lower concentrations of 4AP (5 or 30 mM) or BIC (0.005, 0.5, and 1 mM) led to smaller and shorter-lasting concentration-dependent ECoG power density elevations but did not trigger SD ($n = 9$ total). In most cases, SDs recurred singly or in clusters (Fig. 3B); all recurrent SDs appeared de novo rather than as reentrant or circling SDs on IOS imaging. The frequency of SDs was $0.89 \pm 0.22$/h, $0.26 \pm 0.17$/h, and $0.74 \pm 0.74$/h after 4AP, PG, and BIC, respectively. The overall frequency of SDs decreased over time (only 4AP shown; Fig. 3C) associated with a gradual weakening of seizure power (Fig. 1). Full-field IOS imaging showed that although most SDs originated in the vicinity of the seizure focus, some emerged as far away as the frontal cortex, suggesting that they were triggered by generalized seizure activity (Fig. 3D). Vehicle application did not result in SD.

A proportion of SDs failed to fully penetrate the center of the seizure focus, as previously reported with repetitive ictal events[11]. These non-penetrant SDs were detected mainly as smaller positive DC shifts (Fig. 3E). The non-penetrant SD proportions for the first and the subsequent SDs were 48% and 50%, respectively, after 4AP, and 33% and 67%, respectively, after PG. In contrast, all SDs penetrated the seizure focus during BIC-induced seizures. The non-penetrant SD rate appeared to increase over time suggesting a growing resistance at the focus (Fig. 3F).

**Inhibition by tetrodotoxin confirms seizures as the immediate trigger for SD.** To test whether SDs that developed during a seizure is indeed triggered by seizure activity itself rather than by a direct SD-inducing effect of 4AP, we topically pretreated the cortex with the voltage-gated $Na^+$ channel blocker tetrodotoxin (TTX, 30 μM) 30 min before 4AP application (Supplementary Fig. 3; setup as in Supplementary Fig. 1B, b). TTX depressed ECoG power, blocked seizure induction after 4AP (compare with Fig. 1B), and completely prevented SD occurrence ($n = 5$; $p = 0.005$ 4AP + TTX vs. 4AP alone, $\chi^2$). Importantly, at the end of the experiments, we confirmed that TTX did not directly block SD by successfully inducing an SD via topical KCl application at the remote site and detecting the characteristic negative DC

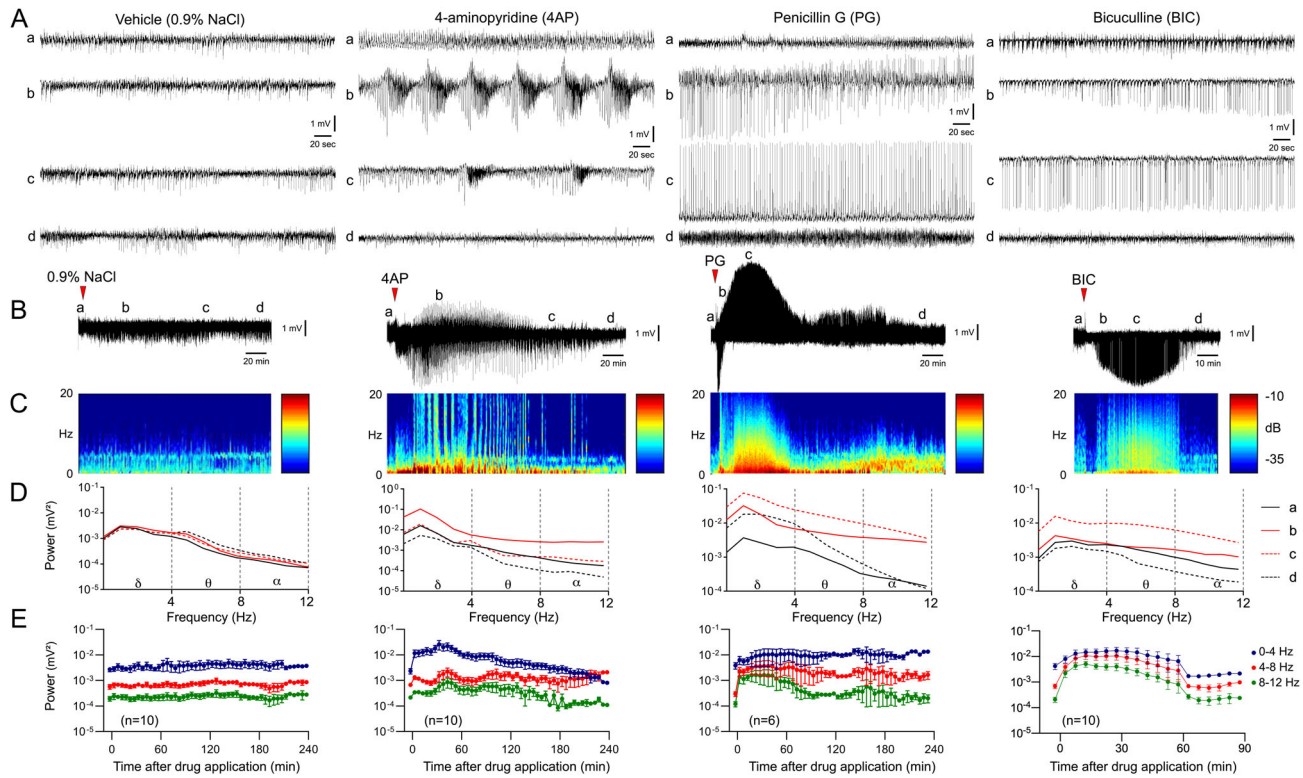

**Fig. 1 Characteristics of focal cortical seizures induced by 4AP, PG, and BIC. A** Representative ECoG tracings show baseline (a), early and late seizure activity (b, c), and resolution (d) during 4AP, PG, or BIC microseizures, as indicated on the full ECoG timeline (**B**). **C** Time–frequency-power spectra of the full timeline shown in (**B**) calculated using Thomson's multitaper method. **D** Power density at the four phases (a-d) computed using FFT. **E** Average ECoG power time course in δ (0–4 Hz), θ (4–8 Hz), and α (8–12 Hz) frequency bands are shown on a logarithmic scale (±SEM). Sample sizes are indicated on the graphs.

potential shift at the TTX application site (Supplementary Fig. 3, upper right panel).

**Power density increase in the δ frequency range predicts SD occurrence.** We next sought to elucidate the determinants of SD occurrence during focal seizures. Comparison of peak power increase in δ (0–4 Hz), θ (4–8 Hz) and α (8–12 Hz) bands among the three epileptogenic agents showed that δ band power increase was greatest after 4AP (Supplementary Fig. 4A), where SD occurrence rate was also the highest (Fig. 3A). Comparison of ECoG power immediately preceding an SD in the δ, θ and α bands between experiments with or without a spontaneous SD further supported an association between power increase in δ range and SD occurrence (Supplementary Fig. 4B). Of note, arterial blood pressure, pH, $pCO_2$, $pO_2$, and glucose remained within physiological ranges and did not predict SD occurrence in any of the experiments (Supplementary Table 1).

**Generalization of seizures across the cortex predicts SD occurrence.** In a subset of mice, 4AP-induced epileptiform activity appeared at the remote ipsilateral anterior recording site and the contralateral homotopic recording sites (36% and 26% of animals, respectively) suggesting seizure generalization (Supplementary Fig. 5, representative tracings a-c and representative spectrograms d-e). We, therefore, examined the relationship between generalization and spontaneous SD occurrence in the 4AP cohort. All nine mice that showed electrophysiological evidence of seizure generalization at the remote sites developed an SD, whereas none of the eight mice that never developed an

SD showed such generalization ($p = 0.031$, $\chi^2$), suggesting that seizure generalization predicts SD occurrence.

To better examine the relationship between seizure generalization and SDs, we determined the spatial reach of seizures across the cortex by making use of the cortical hyperemic response to seizure activity detected with IOS imaging. The periodic bursts induced by 4AP created conspicuous hyperemic transients on IOS images[12] that were locally tightly coupled to epileptic bursts (Supplementary Movie 1; Fig. 4A). Using this surrogate, we tested whether the spatial extent of cortical seizures predicted SD occurrence by placing five regions of interest (ROIs 1–5) 1 mm apart on a line of interest (LOI) extending anteriorly (Fig. 4B). This approach allowed us to visualize the spatial reach of hyperemic transients caused by the periodic burst pattern both along with the LOI and within each ROI over time (Fig. 4C). We found that 69% of all animals ($n = 32$) showed hyperemic transients at the 4AP application site (Fig. 4D, ROI 1); the rest of the cohort (31%) did not show IOS transients despite the presence of electrographic seizures. The fraction of animals showing hyperemic transients in each ROI decreased as a function of distance such that only 28% of animals showed hyperemic transients farthest from the 4AP application site (i.e., ROI 5). Moreover, the latency to the onset of hyperemic transients in each ROI increased as a function of distance, suggesting that the spatial extent of cortical involvement with seizure activity gradually increased over time. When we segregated the experiments based on SD occurrence, we found that focal cortical seizures spread significantly farther from the 4AP application site in experiments where SDs occurred compared with those without an SD ($p = 0.013$; Mann–Whitney test; Fig. 4E). The power map of CBV fluctuations linked to

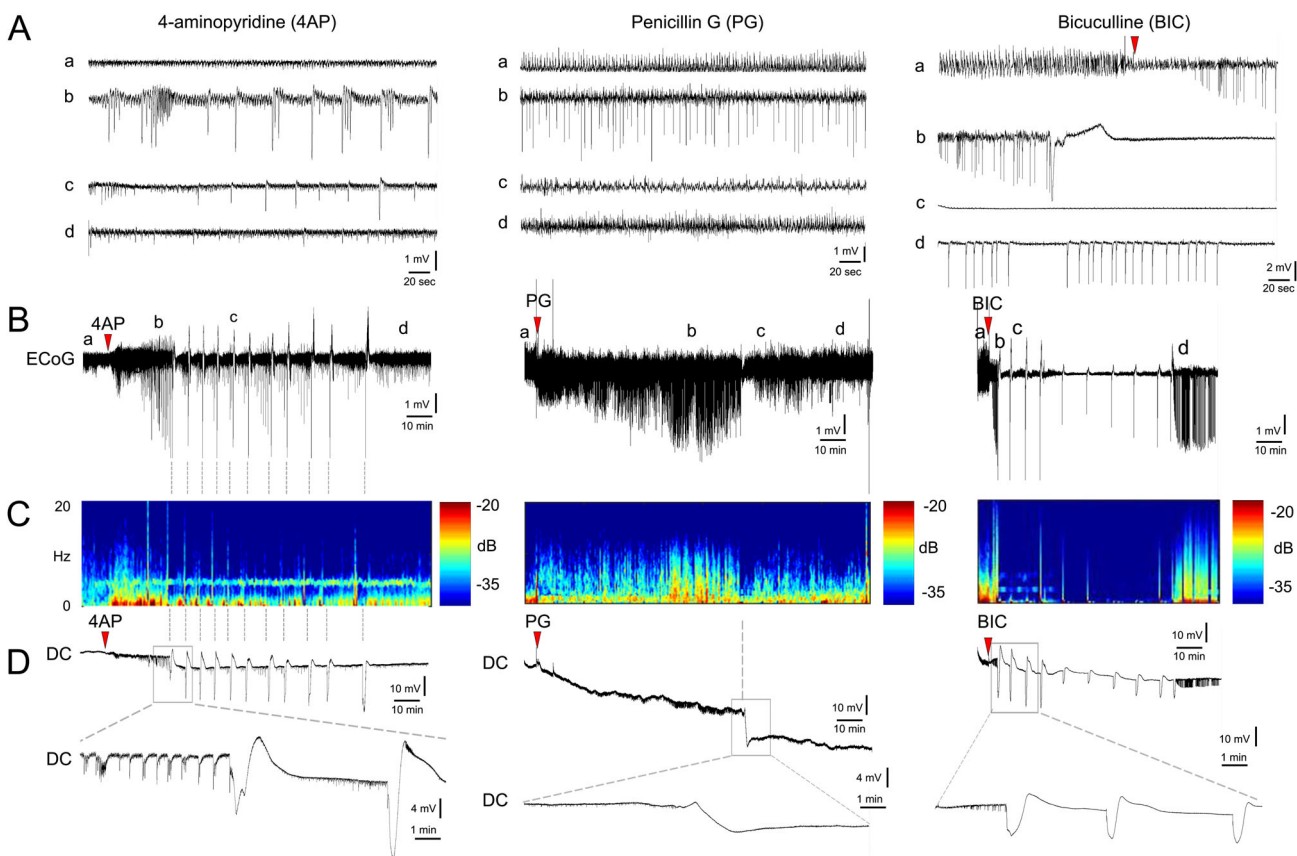

**Fig. 2 Focal cortical seizures trigger spreading depression. A** Representative tracings show typical ECoG phases (a–d) of 4AP, PG, or BIC microseizures, as indicated on the full ECoG timeline (**B**). Red arrowheads show the time of drug application. Vertical dashed lines indicate SDs. **C** Time–frequency-power spectra of the full timeline shown in **B** calculated using Thomson's multitaper method. **D** Corresponding DC tracings show one or more SDs triggered during the seizures as large negative slow potential shifts. Boxes indicate expanded views. These were simultaneously confirmed on IOS as SD waves. A total of 11 spontaneous SDs originated from the 4AP window starting 23 min after drug application in this animal. PG seizure triggered 1 SD at 134 min and BIC seizure triggered 10 SDs between 3 and 60 min.

cortical bursts confirmed this finding (Fig. 4F for representative maps). These data showed that the wider the spatial reach of cortical seizure activity, the higher the likelihood of spontaneous SD occurrence. Importantly, IOS changes during seizure-induced SDs (see Supplementary Fig. 6 for a representative experiment) were similar to hemodynamic changes during SDs induced by other methods (e.g., topical KCl) in migraine models[13,14]. When an SD did not penetrate the seizure focus, it did not cause major IOS changes.

**SD terminates focal cortical seizures and limits their reemergence.** With all three epileptogenic agents, a highly conspicuous effect of SD was seizure suppression that lasted far beyond the characteristic ECoG depression after SD. To systematically test this, we induced SD by topical KCl application at a remote site after seizure onset and quantified its effect on local seizure activity (Fig. 5, setup as in Supplementary Fig. 1B, b). Induced SDs that propagated to and penetrated the seizure focus immediately extinguished all epileptiform activity triggered by 4AP, PG, or BIC. In the wake of SD, ECoG recovered within 10 min but never returned to power levels associated with the seizure activity prior to SD (Fig. 5E). In most cases, seizures did not reignite even when occasional spike activity returned, despite the continued presence of the seizure-inducing agent on the cortex.

Moreover, an SD induced shortly before the application of the seizure-inducing agent diminished the emergence of seizure activity with all three drugs (Fig. 6A). Even when induced 30 min

prior to the 4AP application, SD was still capable of depressing seizure intensity for at least 60 min, suggesting that the antiseizure effect of SD lasts more than 90 min (Fig. 6B). These data showed that SD is a powerful and lasting inhibitor of seizure activity.

**SD limits the generalization of seizures across the cortex.** To determine the effect of induced SD on the spatial reach of seizures across the cortex, we once again used the hyperemia coupled to seizure activity on IOS imaging (Fig. 7). As expected from diminished seizure activity at the focus, an SD induced 30 or 5 min before, or approximately 30 min after 4AP, all decreased seizure generalization assessed by the proportion of distant ROIs showing seizure-induced hyperemia, as well as the average distance from the drug application site (ROI 1) to which the hyperemia reached (Fig. 7B–D, red circles; $p = 0.006$ for 4AP, $p = 0.010$ for PG, $p = 0.016$ for BIC data set; one-way ANOVA).

We also confirmed this by ECoG recordings outside the 4AP site (Supplementary Fig. 7). An SD induced after the generalization of seizure activity to this location strongly suppressed the seizure. The ECoG power returned to the pre-seizure baseline within 10 min but did not reach pre-SD seizure levels for at least 30 min ($p < 0.001$, one-way ANOVA; Supplementary Fig. 7C). These data suggested that SD also limits seizure generalization. Hence, not only larger areas of the neocortex recruited into the seizure increased the chance of SD occurrence (see Fig. 4E above), but SDs, in turn, diminished the area of neocortex recruited into

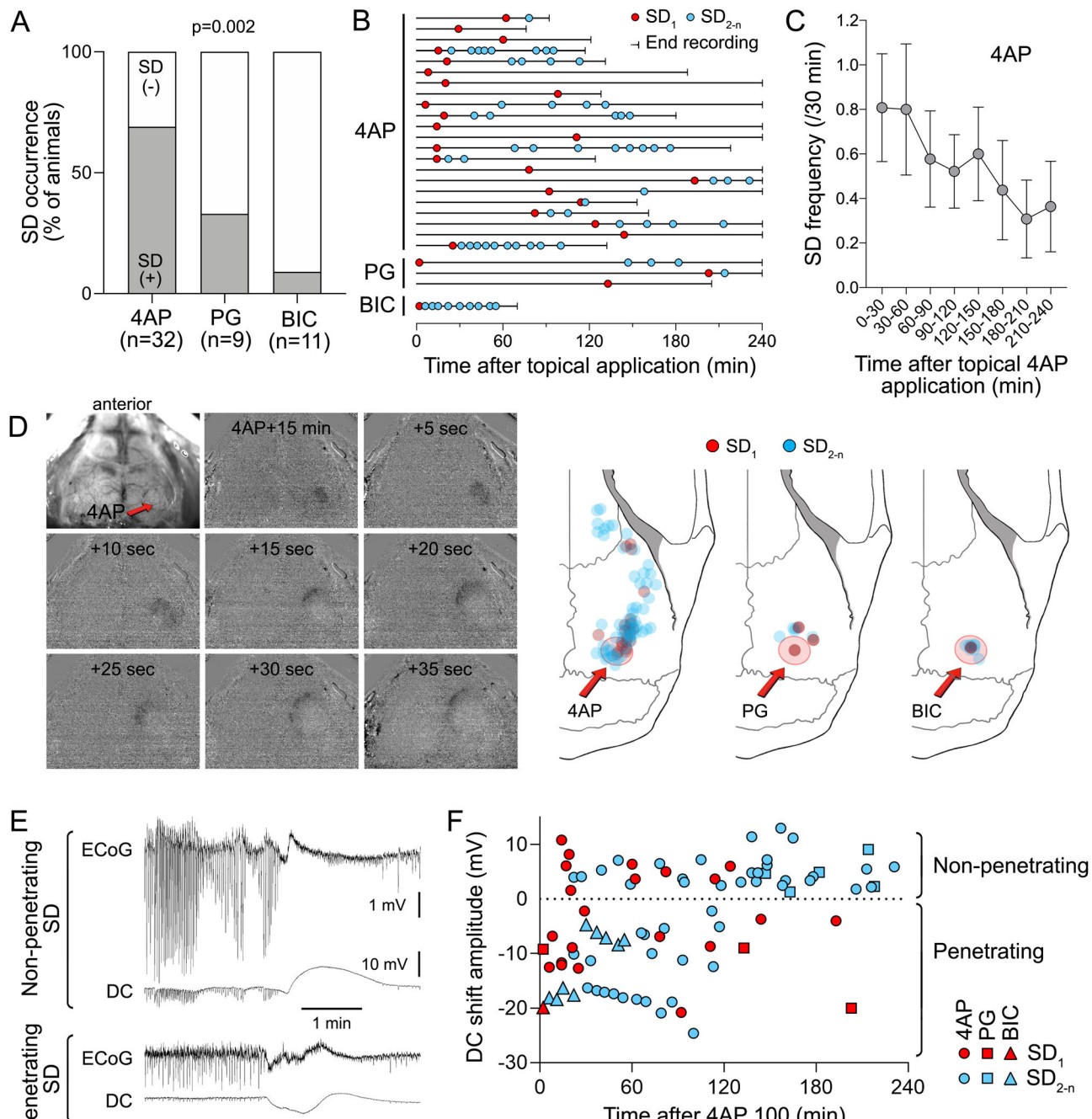

**Fig. 3 Characteristics of SDs triggered by focal cortical seizures. A** The proportion of animals developing SD with each epileptogenic agent ($p = 0.002$; $\chi^2$). Sample sizes are indicated below each bar. **B** Experimental timelines showing first ($SD_1$, red symbols) and subsequent SDs ($SD_{2-n}$, blue symbols). Each line represents the duration of recording in one experiment and each symbol represents one SD. Only experiments with at least one SD are shown. **C** The decrease in SD occurrence overtime after 4AP application (30-min bins) likely reflects weakening seizures (±SEM). **D** Time-lapse IOS subtraction images of an SD originating and spreading anteriorly 15 min after 4AP application (red arrow). The right panel shows the origins of the first and subsequent SDs for each drug. Most SDs originated in the vicinity of the seizure focus. In 4AP seizures, subsequent SDs were more likely to erupt farther from the seizure focus, presumably due to generalization. **E** Representative ECoG and DC tracings of penetrating and non-penetrating SDs recorded by the epidural electrode over the seizure focus. Seizure-induced SDs were uncharacteristically variable in amplitude and duration. **F** The slow (DC) potential shift of the first and subsequent SDs transformed from predominantly negative to predominantly positive over time as the seizure focus became more resistant to SD penetration. Each drug is represented by a different symbol.

the seizure, displaying a feedback mechanism to limit seizure generalization.

**Optogenetically-induced SDs also show the strong antiseizure effect.** Because topical KCl application is an invasive method requiring a craniotomy to trigger SD, we next tested whether

optogenetic SDs induced non-invasively through the intact skull in ChR2+ mice[15] also suppress seizures. We found that a single optogenetic SD induced approximately 30 min after 4AP application indeed extinguished the focal seizure (Supplementary Fig. 8A) and tended to reduce its spatial reach (Supplementary Fig. 8B), non-invasively confirming the antiseizure effect of SD.

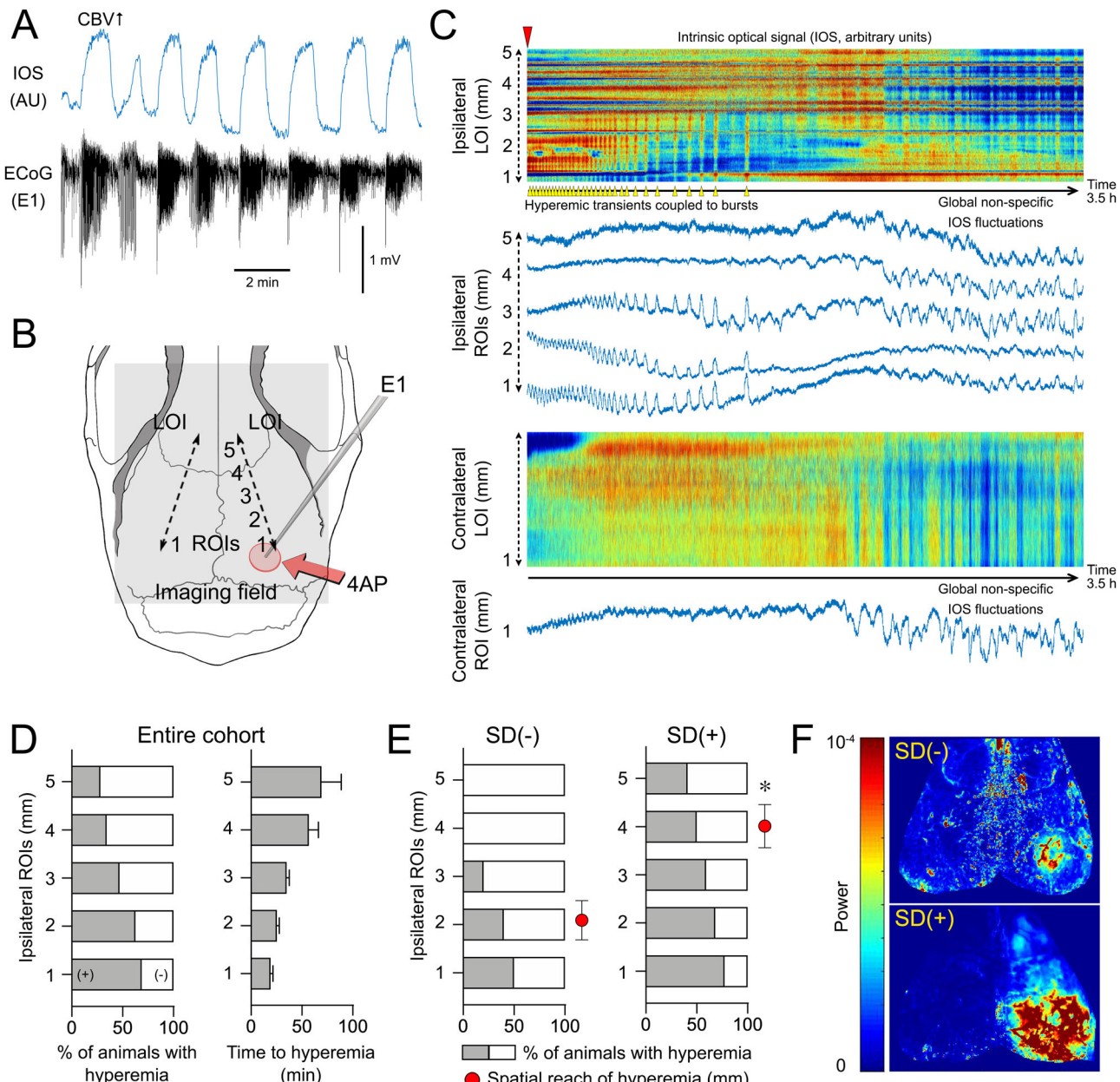

**Fig. 4 Spatial reach of seizures across the cortex predicts SD occurrence. A** Representative 4AP experiment shows hyperemia coupled to seizure bursts (AU, arbitrary unit) used as a surrogate to examine the spatial spread of the seizure. **B** We placed a line of interest (LOI) extending from the drug application site anteriorly on the ipsilateral hemisphere, and one symmetrically on the contralateral hemisphere (dashed lines). Five ROIs (1–5) were placed along with the ipsilateral LOI at 1 mm intervals and a contralateral ROI was placed symmetrically to ipsilateral ROI 1. **C** IOS intensity along with the two LOIs and in each ROI are plotted over time. Red indicates an increase and blue decreases in CBV. Hyperemic transients (yellow arrowheads) reached ROI 3 within 25 min after 4AP (red arrowhead) but did not extend to ROI 4 (experiment without SD). Contralateral LOI and ROI did not show hyperemic bursts, indicating a lack of generalization. Non-specific CBV fluctuations (e.g., due to blood pressure), were distinguishable by their global nature and time course. **D** The proportion of animals that developed hyperemic transients within each ROI and the time of their onset after 4AP application (±SEM). **E** Hyperemic transients showed a farther spread in animals that eventually developed an SD. Red circles indicate the average distance hyperemic transients reached from the drug application site in each subgroup (*p = 0.012, unpaired t-test; ±SEM). **F** Representative CBV power maps show a larger area of cortex with the hyperemic transients in the animal that eventually developed an SD (red high and blue low power).

**Suppression of spontaneous SD occurrence enhances seizure intensity and generalization**. Data thus far strongly suggested that KCl-induced or optogenetic SDs exerted a potent antiseizure effect. We also examined the effect of spontaneous SDs on seizure power and found a significant suppression (Fig. 8). Because the timing of spontaneous SDs was variable among the experiments (unlike induced SDs as above), we only examined the first SD in each experiment. Next, we pharmacologically blocked spontaneous SDs with NMDA receptor blocker MK-801 to test whether eliminating the SDs and thus their antiseizure effect, facilitates 4AP seizures and their generalization. MK-801 decreased the proportion of mice that developed an SD (33%), and reduced SD frequency by more than 80% (0.14 ± 0.06/h, n = 12 mice) compared with control experiments (0.89 ± 0.22/h, n = 32 mice), without suppressing the seizure intensity. On the contrary, SD-suppression by MK-801 facilitated seizure generalization to the

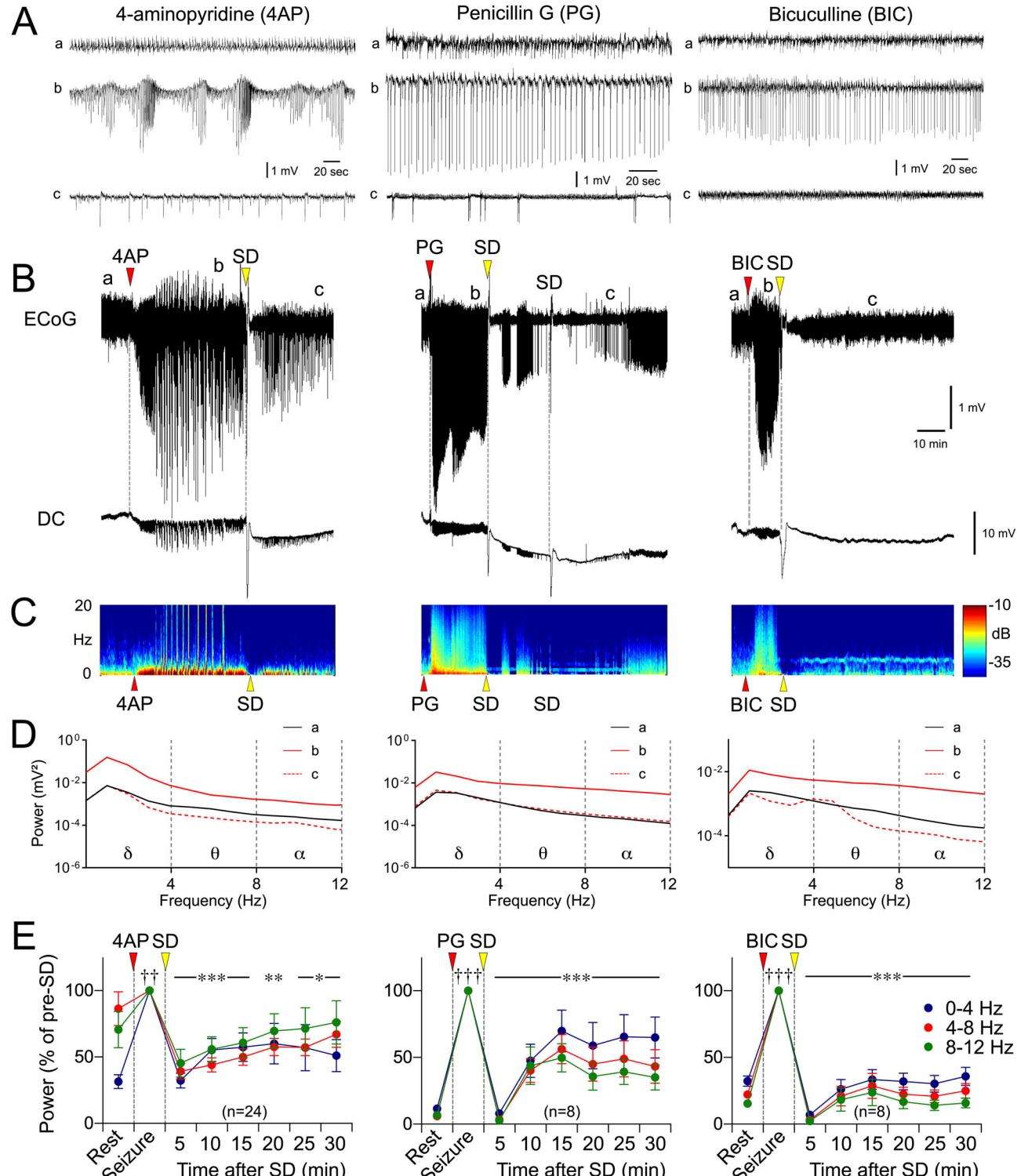

**Fig. 5 Exogenously induced SD terminates focal cortical seizures and limits their reemergence. A** Representative tracings show ECoG at baseline (a), during 4AP, PG, or BIC microseizures (b), and after an exogenously induced SD (c), as indicated on the full ECoG timeline (**B**). Recordings are from the drug application site, (E1, Supplementary Fig. 1B, b). Red arrowheads show the time of drug application. Induced SDs (yellow arrowheads) and the abrupt seizure termination are seen on the DC and ECoG tracings, respectively. Note that a second SD spontaneously originated from the seizure focus in this PG experiment. **C** Time–frequency-power spectra of the full timeline shown in **B** calculated using Thomson's multitaper method also show an abrupt reduction in seizure power associated with SDs. **D** Power density at the three phases (a–c) computed using FFT further demonstrates the lasting seizure power reduction after SD. **E** Average ECoG power time course in δ (0–4 Hz), θ (4–8 Hz), and α (8–12 Hz) frequency bands show ECoG power at rest, during the seizure (5 min prior to SD, and every 5 min thereafter (5–30 min), expressed as % of pre-SD seizure power. All frequency bands show a significant reduction in seizure power for at least 30 min after SD. SD was induced at 37 ± 3 min (13–69) after 4AP application, 19 ± 0 min (17–20) after PG application, and 13±2 min (10–18) after BIC application (mean ± standard error and full range). Sample sizes are shown on the graphs. †-†††$p < 0.05$-0.001 vs. R; *-***$p < 0.05$-0.001 vs. S (two-way ANOVA for repeated measures; ±SEM).

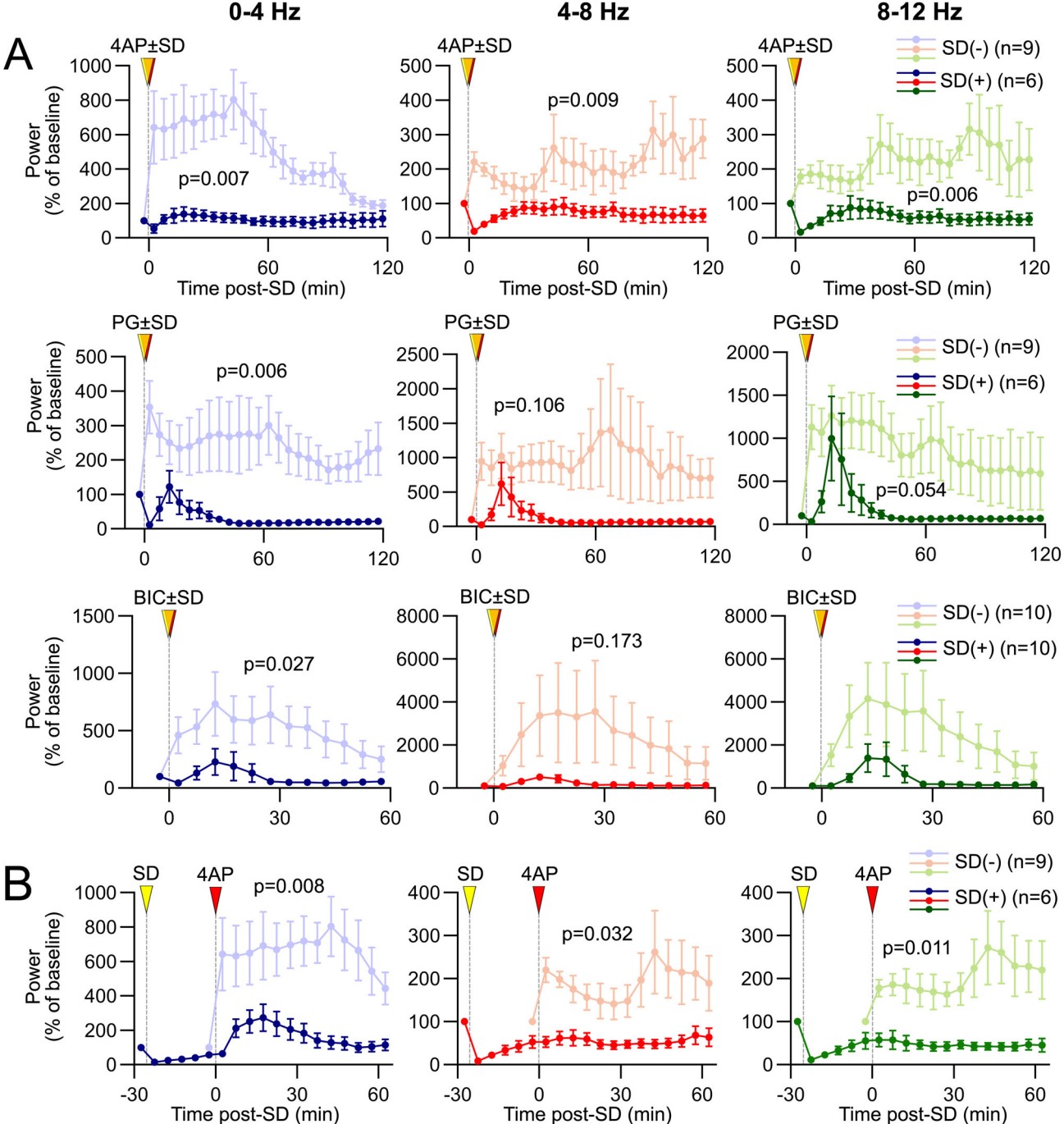

**Fig. 6 Pre-treatment with SD strongly inhibits the emergence of seizure activity. A** Average ECoG power time course graphs (5-min bins) expressed as % of baseline (−5 min) show that an SD induced (yellow arrowheads) 5 min before 4AP, PG or BIC application (red arrowheads) significantly diminished seizure intensity (bold curves) compared with controls without an SD (faint curves). **B** An SD induced 30 min before 4AP application still significantly suppressed subsequent seizure activity for at least 90 min. Data are from the drug application site (E1, Supplementary Fig. 1B). Each frequency band is shown separately. Exact p values are shown for SD (−) vs. SD (+), using two-way repeated-measures ANOVA.

ipsilateral and the contralateral remote recording sites (92% and 89%, respectively; Fig. 9A), compared with control animals (see above). Indeed, ECoG power increase after 4AP was significantly larger in both the ipsilateral and the contralateral remote recording sites in the MK-801 group (Fig. 9B). Once again using the hyperemic transients coupled to periodic bursts, we further confirmed robust seizure generalization in both the ipsilateral and the contralateral ROIs after MK-801 on IOS images (Fig. 9C). Altogether, these data indicated that spontaneously occurring SDs

during highly focal seizures are capable of exerting an antiseizure effect. Suppression of SD occurrence by MK-801 eliminated this endogenous antiseizure mechanism.

**Familial hemiplegic migraine knock-in mice.** We next sought to reproduce our findings in a migraine-relevant knock-in mouse model expressing the S218L familial hemiplegic migraine type 1 (FHM1) mutation in the Cav2.1 voltage-gated Ca2+ channels.

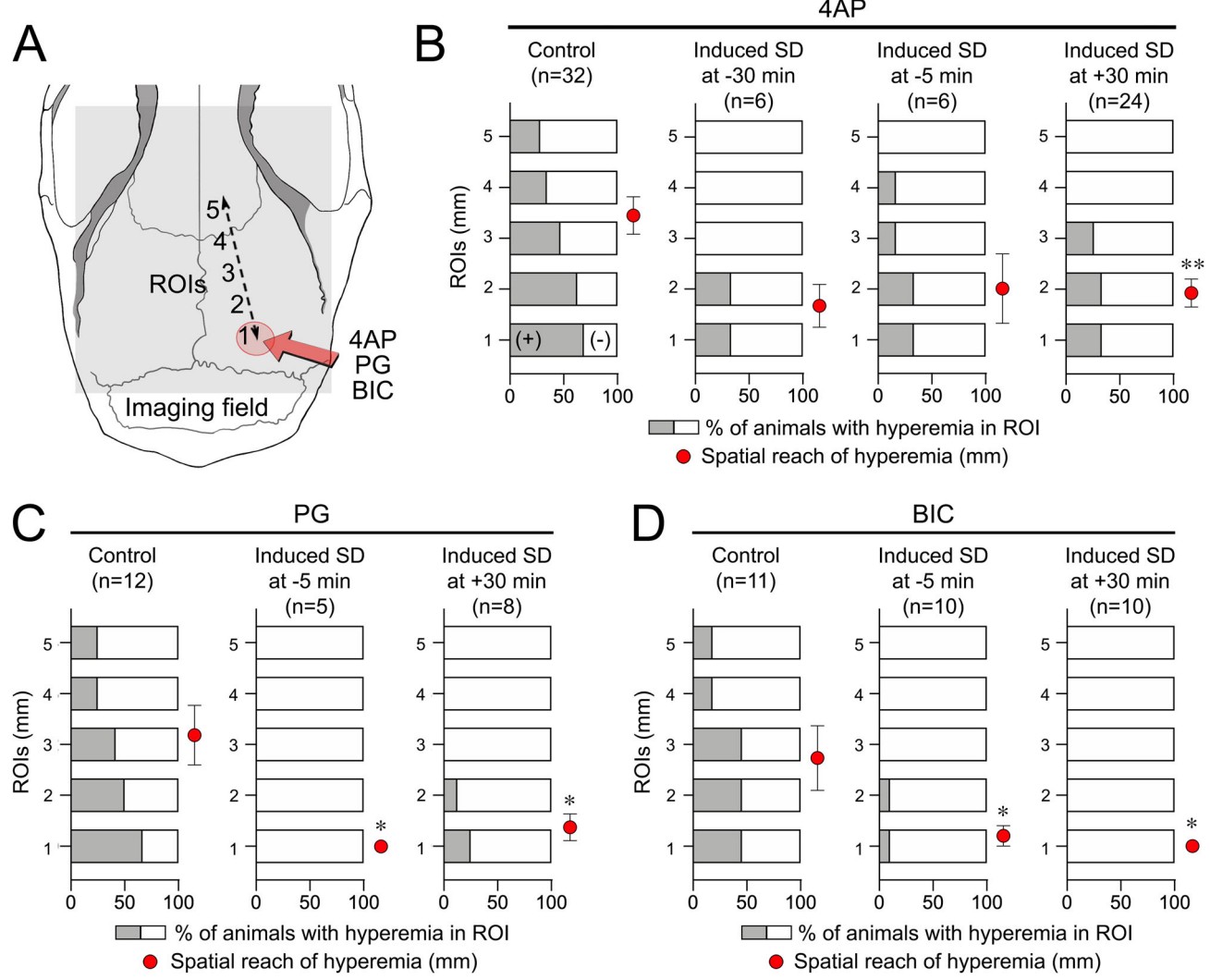

**Fig. 7 SD limits the spatial reach of seizures across the cortex. A** Diagram showing the ROI placement for IOS analysis. **B–D** The proportion of animals that developed hyperemia within each of the five ROIs coupled to 4AP, PG, or BIC seizures show that the spatial reach of seizures was diminished by an SD induced 30 min before (−30 min), 5 min before (−5 min) or 30 min after (+30 min) drug application. All SDs were remotely induced by topical KCl application using the setup shown in Supplementary Fig. 1B, b. The control group did not receive any SD intervention from outside, but 22 animals in 4AP, 3 animals in PG, and 1 animal in the BIC control group developed spontaneous SDs (as summarized in Fig. 3A, B). Red circles indicate the average distance hyperemic transients reached from the drug application site in each subgroup (no hyperemic transients in any ROI is 0 mm). *-**p < 0.05-0.01 vs. control, one-way ANOVA followed by Dunnett's multiple comparison's tests (±SEM).

Topical application of 4AP (100 mM) induced typical seizures in both the FHM1 knock-in mice and their wild-type (WT) littermates (Supplementary Fig. 9). Despite lower peak power levels reached during the seizures, 7 out of 8 FHM1 mutants developed SDs, compared with only 1 out of 6 of their WT littermates ($p = 0.008$, $\chi^2$; Supplementary Fig. 9B). In FHM1 mutants, SDs often occurred in clusters (Supplementary Fig. 9C). The average SD frequency in FHM1 mutants ($3.0 \pm 0.7$ SDs/h) was markedly higher than the WT littermate that developed an SD (1.3 SDs/h), as well as the CD1 mice ($0.9 \pm 0.2$ SDs/h). The majority of SDs in FHM1 mutants penetrated the seizure focus and thus were associated with negative DC potentials shifts. However, the DC shift amplitude decreased over time suggesting a gradually developing resistance to SD at the seizure focus similar to CD1 mice (Supplementary Fig. 9D). More importantly, spontaneously developing SDs in FHM1 mutants also suppressed seizure intensity (Supplementary Fig. 9E).

**Generalized seizures upon systemic kainate administration.** Lastly, we extended our findings to a generalized seizure model using systemic kainate injection (60 mg/kg, intraperitoneal in CD1 mice, n = 7). Electrophysiological recordings from the hippocampal CA1 region revealed a characteristic pattern of recurrent seizures consistent with electrographic status epilepticus in all mice that started gradually and gained strength within 30–60 min of kainate administration. Seizures were characterized by rhythmic 15–30 Hz activity with evolution in amplitude to 300% of baseline followed by the onset of 1–2 Hz rhythmic discharges measuring up to 8 mV. Discharges slowed and decreased in amplitude after 15–60 s giving way to post-ictal voltage attenuation. These recurred every 1–3 min with increasing intervals (Fig. 10A). All mice developed a hippocampal SD during kainate seizures. SDs occurred either during a seizure burst or shortly thereafter, the latter suggesting an SD origin farther from the recording site. In the wake of an SD, seizures were completely

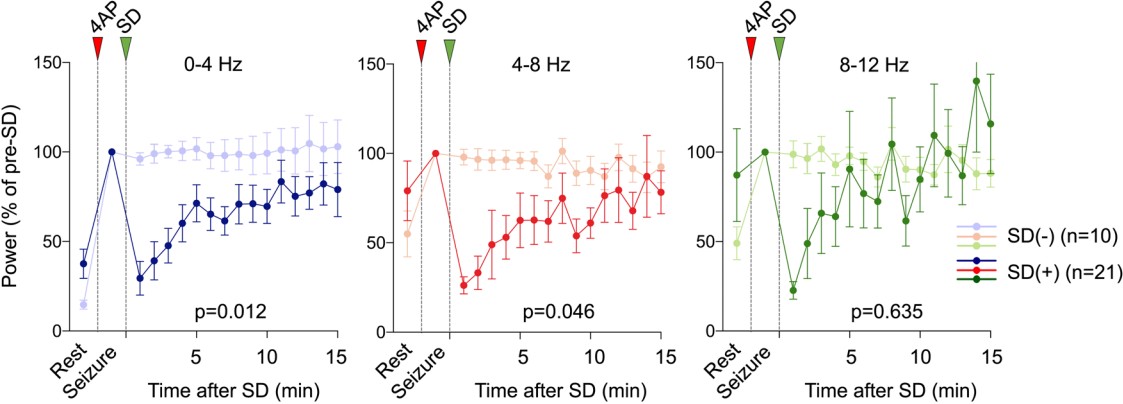

**Fig. 8 Spontaneously occurring SD terminates focal cortical seizures and limits their reemergence.** Average ECoG power time course in δ (0–4 Hz), θ (4–8 Hz), and α (8–12 Hz) frequency bands show ECoG power at rest, during the seizure (5 min prior to SD, and every minute thereafter, expressed as % of pre-SD seizure power. Spontaneously arising SD significantly diminished seizure intensity (bold curves) compared with controls without an SD (faint curves) in δ (0–4 Hz) and θ (4–8 Hz) bands (two-way ANOVA for repeated measures; ±SEM). Sample sizes are shown on the graph. Only the first SD in each experiment is analyzed.

suppressed for 10 min, and seizure power did not return to pre-SD levels for at least 30 min (Fig. 10B).

## Discussion

Our systematic examination using topical application of three different seizure-inducing agents shows that highly focal cortical seizures are capable of triggering SD. Both the ECoG's greater power increase in the δ frequency band and greater spatial extent of generalization were positive predictors of SD occurrence, which was also enhanced by a human migraine mutation. In turn, SDs suppress seizures for more than 30 min, far beyond the brief electrophysiological silence associated with SD. Generalized seizures upon systemic kainate administration reproduced nearly identical findings.

**A unifying theory.** Based on these findings, we propose a unifying theory wherein SD is a fundamental endogenous antiseizure mechanism in the central nervous system. As such, SD is triggered if and when synchronized focal neuronal network activity collectively raises extracellular $K^+$ above the 12 mM threshold in a minimum critical volume of tissue, which is estimated to be ~1 mm³[16]. SD then acts as an 'emergency brake' or 'reboot' extinguishing the seizure and propagates centimeters away from the focus to exert a broader antiseizure effect. The latter is clinically perceived as a migraine aura (Supplementary Movie 2). Clearly, the organism is better off experiencing a migraine aura than seizures, status epilepticus, or generalized convulsions in terms of survival advantage, because they can lead to loss of consciousness, bodily harm, and prolonged post-ictal encephalopathy, and may even be terminal. Therefore, SD can be beneficial in the context of seizures. As such, our theory may explain the evolutionary purpose and persistence of SD.

Our theory can also explain the clinical overlap between epilepsy and migraine with aura, two comorbid, chronic-episodic, paroxysmal disorders. As eloquently outlined by Rogawski[17], epilepsy, and migraine with aura share a number of common features: (a) higher prevalence of epilepsy in migraine with aura sufferers than the general population and vice versa[18,19]; (b) hyperexcitability in both migraine with aura and epilepsy[20,21]; (c) strong genetic underpinnings (e.g., *CACNA1A, ATP1A2, SCN1A, NOTCH3,* and *PRRT2*) associated with both migraine with aura and epilepsy[17,22,23]; (d) frequent occurrence of postictal migrainous headaches after seizures[24,25]; and (e) efficacy of antiepileptic drugs, such as valproate and topiramate, in migraine prophylaxis.

All these shared features support the notion that microseizures may serve as triggers for migraine aura, i.e., SD. Of course, it unlikely that all auras are triggered by focal seizures and other trigger mechanisms exist.

**Relevance for microseizures.** Given their average spatial reach of only a few millimeters, the focal epileptiform activity generated in our model was akin to human microseizures that are typically below the spatial detection limit of conventional EEG and only detected using implanted microelectrodes in epilepsy patients, as well as in normal subjects[26–28]. Such localized, subclinical epileptiform activity can in fact precede clinical seizures by many hours[29]. Microseizures can spread by recruiting neighboring tissue, and as such, act as a precursor or seed for clinically overt seizure activity[30]. Moreover, microseizures can lead to potentiation of synapses and kindling over time and may be considered an intermediate stage in epileptogenesis[31]. Given the high rate of SD occurrence we detected under a general anesthetic regimen known to suppress SD susceptibility[32], seizures may trigger SD even more frequently in the unanesthetized brain. Once triggered, SD propagates regionally to prevent seizure spread, and perhaps even kindling, thereby exerting a lasting antiepileptic effect. Most importantly, seizures can perpetuate and lead to excitotoxic cell death, whereas in the absence of metabolic compromise, SD is fundamentally self-limiting (typically < 1 min) and non-injurious even after numerous events[33], except when prolonged, under hypoxic/ischemic conditions or involving the brainstem [1,34,35].

**Evidence from other model systems.** Seizures culminating in SD have been observed under various experimental and clinical conditions. For example, focal cortical seizures after topical application of sodium penicillin salt triggered cortical SD waves in rats[36], spontaneous focal seizures in both post-cerebral malaria and tetanus toxin mouse models of epilepsy triggered SDs[37], focal cortical seizures after electrical stimulation or topical application of 4AP triggered cortical and/or brain stem SD and death in mutant mouse models[6,35], and generalized seizures (e.g., maximal electroshock; systemic pentylenetetrazol, tetanus toxin or flurothyl; audiogenic; absence) triggered cortical or subcortical SD in multiple species[37–39]. Moreover, in the acutely injured human brain, subdural recordings showed that more than half of the patients who developed seizures also developed SDs, and in most cases, seizures appeared to trigger SD[8]. These data suggest that SD triggered by seizure activity is a ubiquitous phenomenon.

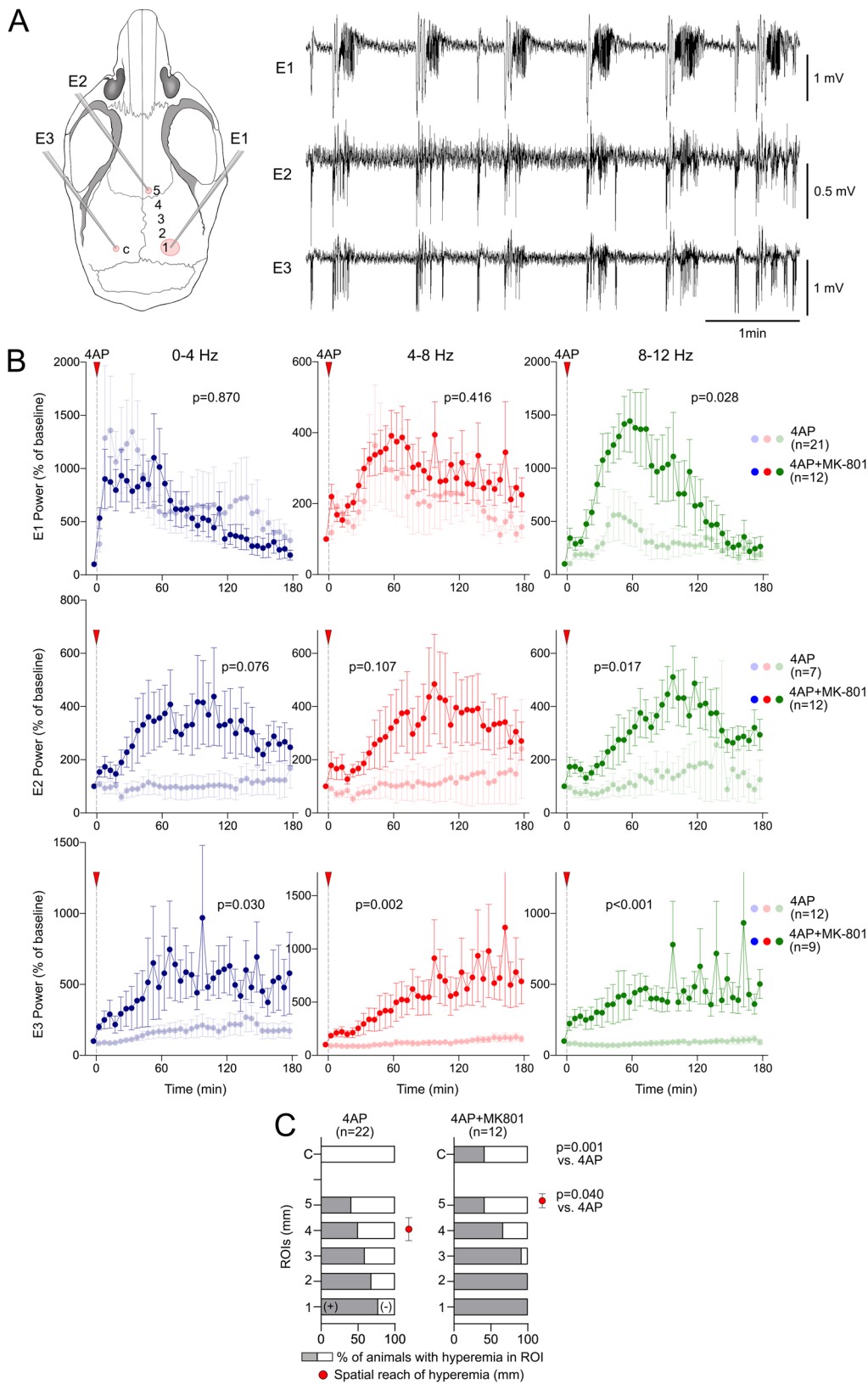

Even recurrent SDs were predicted by the extended Hodgkin–Huxley model[9,10]. More importantly, SDs that developed during focal or generalized seizures were associated with seizure termination, after SD seizures did not return for more than 20 min, and when an SD did not develop seizures typically intensified[38–41]. Similarly, SDs that developed during seizures in the acutely injured human brain was followed by suppression of seizure activity for more than 15 min even when ECoG activity has recovered[8]. Mechanically or KCl-induced SDs also potently extinguished and/or prevented focal and generalized seizures, sometimes for more than 90 min[42–45]. Altogether, the data strongly suggest that SD is a fundamental endogenous

**Fig. 9 Suppression of spontaneous SDs enhances seizure intensity and generalization. A** Left panel: Diagram showing electrode (E1-E3) and ROI (1–5, C) placement. Representative ECoG tracings showing seizure generalization to the remote ipsilateral (E2) and contralateral (E3) electrodes. Such synchronized electrophysiological generalization was common in the MK-801 group but rarely observed in control animals that were allowed to develop SDs. **B** Averaged ECoG power time courses normalized to pre-seizure baseline are shown separately for each electrode and frequency band. Seizure generalization to the remote ipsilateral (E2) and contralateral (E3) electrodes is significantly higher in the MK-801 group (bold symbols) that did not develop any SD compared with time controls that were allowed to develop spontaneous SDs (4AP alone, faint symbols). Seizure intensity at the local electrode (E1) is much less affected (only in 8–12 Hz band). Red arrowheads show the time of the 4AP application. Exact p values are shown on the graphs (two-way ANOVA for repeated measures; ±SEM). **C** The proportion of animals that developed hyperemia within each of the six ROIs coupled to 4AP seizures are shown with and without MK-801 treatment. The proportion of animals that showed hyperemic transients in the contralateral ROI (c) was significantly higher in the MK-801 group compared with controls ($p = 0.001$, $\chi^2$). The average distance hyperemic transients reached from the drug application site was significantly higher in the MK-801 group ($p = 0.040$, $t$-test with Welch's correction; red circles; no hyperemic transients in any ROI = 0; ±SEM).

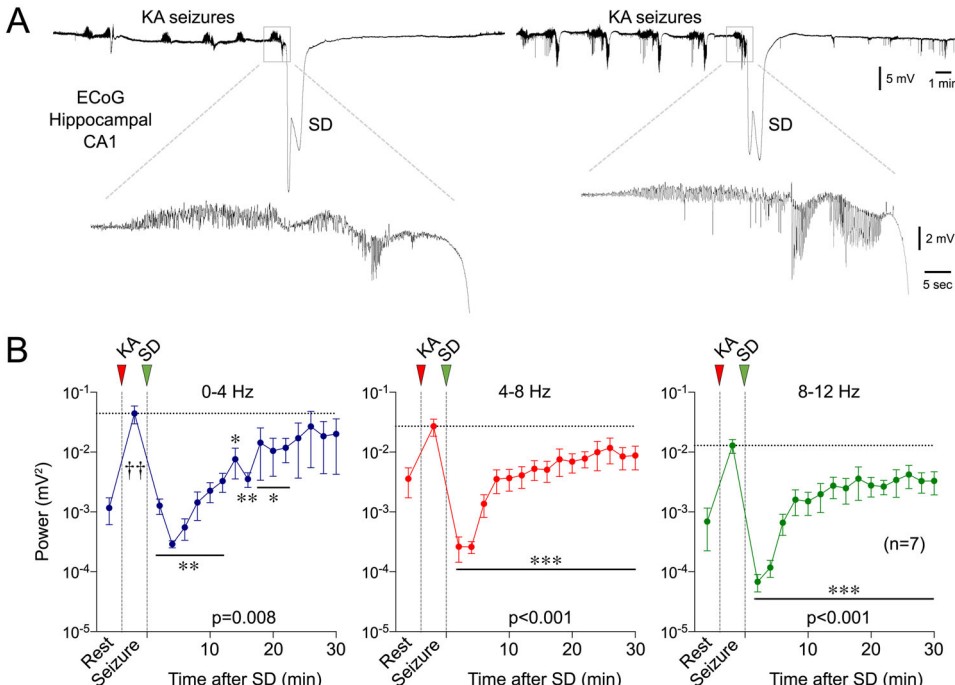

**Fig. 10 Spontaneous SDs during systemic kainic acid-induced global seizures terminate the seizures and limit their reemergence. A** Representative hippocampal recordings (DC) show seizure characteristics and SD emergence in two animals. **B** Average ECoG power time course in δ (0–4 Hz), θ (4–8 Hz), and α (8–12 Hz) frequency bands show ECoG power at rest, during the seizure (5 min prior to SD, and every 2 min thereafter. All frequency bands showed a significant reduction in seizure power for at least 20 min after SD. The sample size is shown on the graph. *-***$p$ < 0.05–0.001 vs. pre-SD seizure power (one-way ANOVA for repeated measures; ±SEM).

antiepileptic mechanism that is recruited when the intensity of network activity reaches a threshold. Indeed, SDs triggered by acute brain insults (e.g., concussion, intracranial hemorrhage) may also serve to suppress focal seizure activity that might be triggered by the insult, or its generalization, providing yet another potential survival advantage for the organism.

The duration of the antiseizure effect appeared to be highly variable among studies, ranging from just a few minutes to more than 90 min. This was likely due to the intensity of the seizure-inducing stimulus or the chronic epilepsy model used in each study. For example, when high concentrations of drugs were topically applied on a wide area of cortex, seizures returned shortly after SD[43], whereas spike-wave discharges in a genetic model of absence epilepsy were suppressed for at least 4 h by a single SD[44]. The latter is clearly a translationally more relevant model than the topical application of potent seizure-inducing drugs. Therefore, we believe in the majority of cases where weak focal epileptiform activity evolves into a generalized seizure relatively infrequently, the antiseizure effect of SD is likely to be efficacious for longer periods of time. Nevertheless, recurrent

seizures also resulted in recurrent SDs in most studies, including ours, which was reminiscent of symptomatic occipital lobe epilepsy presenting with status migrainosus and diffusion-weighted MRI abnormalities consistent with SD[46]. Regardless of the context, however, on the whole, the occurrence of recurrent SDs significantly reduced the seizure burden in experimental models and is likely to do so in the clinical setting as well. It should be noted that chronic epileptic brains become resistant to SD[11,47] and acute but lasting seizures also render the tissue relatively resistant to SD[43]. This was evident in the failure of many SDs to propagate into the seizure core particularly during the later stages of our experiments as well. Indeed, persistent seizure activity in the process of kindling makes seizures less likely to culminate in SD[48].

**Potential mechanisms.** The mechanisms triggering SD during intense neural activity are almost certainly related to elevated extracellular [K$^+$]. It is also possible that SD occurrence in our model simply reflects the deterioration of tissue health, but this is

unlikely as SDs occurred in some cases within mere minutes after seizure onset, at a time when the tissue was robust, healthy, and capable of sustaining more seizures thereafter. The mechanism(s) by which SD exerts its antiseizure effect is more difficult to discern. Profound shifts in extracellular concentrations of most if not all ions, elevated levels of numerous neurotransmitters and neuromodulators, metabolic and hemodynamic alterations, and morphological and gene expression changes, can all contribute to the antiseizure effect to some extent. It has been proposed that the concept of the ictus in epilepsy could be broadened to include a trajectory of seizures terminated by SD, seen in spontaneous epilepsy[7] and predicted by biophysics[9,10]. Given the overlapping cellular and electrophysiological mechanisms of seizures and SD, it is not easy to find an intervention that directly and selectively blocks SD occurrence without affecting the seizures to dissect the causality between these two phenomena and their mechanisms. Any intervention that inhibits one is also likely to directly affect the other. Regardless, data overwhelmingly show that the net effect of SD on seizure activity is a lasting suppression.

**Is SD physiological?** Research into SD over the past two decades, heavily in the context of acute brain injury, has portrayed SD as a pathological depolarization state associated with membrane failure. Both as a consequence of and contributor to brain injury, SD signified harm. However, older studies paint a different picture, where SD could be readily induced by single epileptiform spikes[36,48], and even by sensory activation if the cortex was hyperexcitable[38,49]. The latter is intriguing given that migraine attacks can be triggered by sensory stimuli as well[50]. Intense synaptic activation via long-range inputs could also trigger SD[51,52], and spontaneous SDs have been unequivocally demonstrated in genetically susceptible mice[35]. In fact, SD may be easier to develop than a seizure[53], and recent models suggest seizures and SD are part of a dynamic continuum of neuronal membrane state[9,10]. There are many physiological (i.e., part of normal function, or beneficial) processes that can be pathological when they are overactive or occur at the wrong place and time, and SD might be one of those. When SD occurs in the wrong context (e.g., brain injury), it can have pathological consequences. But even in the injured brain, the interplay of seizures and SD is complex[8,54–56], and seizures can be terminated by SDs[56,57]. In conclusion, SD may be an integral part of the brain function and serve fundamental roles (e.g., antiseizure effect). Further research into the potential beneficial effects of SD may reveal other mechanisms that facilitated evolutionary selection and conservation of SD.

## Methods

All experiments were approved by the Massachusetts General Hospital Institutional Animal Care and Use Committees and followed the NIH Guide for Use and Care of Laboratory Animals (NIH Publication No. 85-23, 1996).

**Animal preparation**. We used wild-type mice ($n = 165$, CD1, male, $3.2 \pm 0.1$ months, Charles River Laboratories, Wilmington, MA, USA), familial hemiplegic migraine type 1 (FHM1) knockin mice heterozygous for the S218L mutation in the mouse *Cacna1a* gene encoding for the $\alpha1_A$ pore-forming subunit of $Ca_V2.1$ voltage-gated calcium channels ($n = 7$ males, 1 female; $8.9 \pm 2.7$ months) and their wild type littermates ($n = 4$ males, 2 females; $8.3 \pm 1.6$ months), and transgenic mice expressing channelrhodopsin-2 (ChR2$^+$, $n = 10$ males and 8 females, $7.5 \pm 0.6$ months, B6.Cg-Tg (Thy1-COP4/EYFP)18Gfng/J, Jackson Laboratories, Bar Harbor, ME, USA)[58,59]. Mice were kept under diurnal lighting conditions, room temperature of approximately 25 °C and relative air humidity of 45–65%. Animals were kept in one cage in groups of two to four. The majority of mice were fasted overnight ($17.4 \pm 0.3$ h; $n = 173$), while a small cohort was studied without fasting ($n = 17$); results did not differ and were pooled. Before anesthesia induction blood glucose was measured via the tail vein (Accu-Chek Guide, Hague Road, Indianapolis, USA). Mice were then anesthetized with isoflurane (3% induction, 1.25% maintenance) and allowed to breathe spontaneously on a mixture of 70% $N_2O$/30% $O_2$. The femoral artery was cannulated for continuous blood pressure (PowerLab; ADInstruments, Colorado Springs, MO, USA),

$pCO_2$, $pO_2$, and pH measurements. Rectal temperature was kept at 36.5–37.0 °C using a thermostatically controlled heating pad (CWE Inc., Ardmore, PA, USA). Mice were then placed on a stereotaxic frame. The skull was exposed, and mineral oil applied to prevent bone drying. Burr holes were drilled under saline cooling and the dura was kept intact.

**Cortical electrophysiological recordings and data processing**. Slow (DC) potential and electrocorticogram (ECoG) were continuously recorded using epidural glass micropipettes filled with 0.9% saline and a differential extracellular amplifier (bandpass 0.3–300 Hz; EX4-400, Dagan Corporation, Minneapolis, MN) and digitized at 1 kHz for offline analyses (PowerLab; ADInstruments, Colorado Springs, MO, USA). An Ag/AgCl reference electrode was placed subcutaneously in the neck. To eliminate non-physiological signals a 20 Hz digital low-pass filter was applied (Labchart 5 Pro, ADInstruments, Colorado Springs, MO, USA). The ECoG power spectrum density was computed using fast Fourier transform (FFT, cosine-bell window, size 1024, window overlap 93.75%) on 5-min bins of continuous noise-free segments (Labchart 8 Pro, ADInstruments, Colorado Springs, MO, USA). These were then averaged for δ (0–4 Hz), θ (4–8 Hz), α (8–12 Hz) bands, and plotted over time. In addition, time-varying spectrograms were computed using Thomson's multitaper method (window length 10 s, window step 1 s, bandwidth 20 Hz, number of tapers 9; Chronux toolbox, http://chronux.org; version 2.12 v03, 2018).

**Hippocampal recordings**. DC potential and ECoG were continuously recorded using a glass micropipette in the hippocampal CA1 region (2 mm posterior and 1.5 mm lateral from bregma at a depth of 1.4 mm). After kainic acid administration isoflurane was gradually lowered from 1.5 to ~0.58 ± 0.08% over 45 min until small-amplitude clonic leg movements appeared as a sign of generalized seizure in the absence of purposeful movements in response to pain.

**Intrinsic optical signal (IOS) imaging**. We simultaneously performed transcranial IOS imaging of cerebral blood volume (CBV) changes to complement electrophysiological recordings in all experiments to visualize the origins and propagation of SDs and the spatial extent of seizure activity as previously reported[12,14,60]. IOS imaging started prior to skull drilling to detect inadvertently triggered SDs during the preparation as an exclusion criterion. We achieved skull translucence using topical mineral oil application and diffusely illuminated the skull surface using a white or 530 nm green LED (M530L3, FB530-10, Thorlabs, Newton, NJ, USA) and an aspheric condenser lens (ACL2520U-DG15-A, Thorlabs, Newton, NJ, USA). We acquired full-field images using a USB camera (1 Hz, 640 × 480 pixels; Yaw-Cam 0.6.2, 2018). The camera and light source was positioned to minimize surface glare. The green channel of all images was processed with an in-house MATLAB (R2018b) code using the modified Beer-Lambert law as previously described[61] to enhance the signal changes reflecting total hemoglobin, and thus CBV. Images were downsampled to 320 × 240 pixels and the reflected light intensity of each pixel in each frame over the whole recording time was calculated. Two lines of interest (LOIs) were placed, one between the drug application site (hereafter referred to as the *focus*) and the anterior recording site, and one symmetrically on the contralateral hemisphere. In addition, regions of interest (ROIs) were placed along with the LOI at 1-mm intervals, and full-field band power maps were generated, to quantify regional seizure spread.

**Drugs**. We applied 4-aminopyridine (4AP; 5, 30, or 100 mM; Sigma-Aldrich), penicillin G (PG; 188434 IU/ml, sodium salt; Sigma-Aldrich, St. Louis, MO, USA), 1(S),9(R)-(−)-bicuculline methiodide (BIC; 5 mM; 1 mM, 0.5 mM, 0.05 mM, and 5 μM; Sigma-Aldrich) or vehicle (0.9% saline) topically and repeated it once 10 min later. Cortex was not washed afterwards. In a subset of experiments, tetrodotoxin (TTX; 30 μm in sodium citrate buffer, pH 4.8; Enzo Life Sciences, Farmingdale, NY, USA) was topically applied 30 min prior to 4AP. MK-801 (1 mg/kg, $n = 7$; 3 mg/kg, $n = 5$; Sigma-Aldrich) and KA (60 mg/kg; Sigma-Aldrich) were administered intraperitoneally. Topical concentrations and systemic doses were selected based on the literature [6,62–64].

**Experimental design and setup**. Electrophysiology and optical imaging were carried out simultaneously in the same anesthetized animal throughout the experiment (Supplementary Fig. 1A). Various electrode, fiberoptic, and burr hole configurations were used depending on the experimental design. Basic configurations used in the majority of experiments are shown in Supplementary Fig. 1B. Variations to this configuration, when present, are indicated on subsequent figures where the relevant data are presented. The maximum duration of recordings was 240 min. We stopped the recordings if seizure activity diminished (<3 spikes/20 s) for at least 10 min suggesting waning of drug effect ($n = 10$). Three experiments (76, 90, and 117 min) were additional controls done at later stages of the project to ensure the stability of the model system, and to record high-quality movies. Only one wild type and two FHM1 mice died during the recordings.

**Rigor and statistics**. Real-time electrophysiological recordings and IOS images precluded blinding. In the absence of a priori experience with the experimental

approach, sample sizes were chosen empirically. Data were analyzed using Prism 8 (GraphPad Software) and expressed as mean and standard error. Statistical tests used for each dataset and the sample sizes are indicated where they are presented in text, on the figures, or in figure legends.

**Reporting summary**. Further information on research design is available in the Nature Research Reporting Summary linked to this article.

## Data availability

The data that support the findings of this study are available from the corresponding author upon reasonable request. Source data are provided with this paper.

## Code availability

The MATLAB code for calculating CBV changes can be accessed through the following link: https://github.com/it078/CBV_Pixel_Analysis_Seizure_Spread.git.

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

## Acknowledgements

This work was funded by NIH (R01NS102969 to C.A., 2R01EB014641 to S.J.S.).

## Author contributions

I.T. conceived the study, designed and performed experiments and data analysis, wrote the manuscript, and drew the figure illustrations. C.A. conceived the study, designed experiments and data analysis, performed circuit reconstruction and wrote the manuscript. D.C. contributed data analysis, experimental design, and revision of the manuscript. A.L. contributed to experimental design. I.L. contributed to revision experiments. T.Q. contributed reagents. A.M., F.S., M.E., and S.J.S. revised the manuscript.

## Competing interests

The authors declare no competing interests.
