## [Peer Review File · Nature Communications]

REVIEWER COMMENTS

Reviewer #1 (Remarks to the Author):

Spreading depression (SD) occurs in neurological disorders such as migraine, Epilepsy and brain injury. Is it just a concomitant phenomenon, or harmful as many studies believe, or protective as this study suggests? In this study, authors used in vivo mouse model of focal seizures induced by 4-AP, PG and BIC both in wild type and transgenic mice, and electrophysiological, IOS imaging and optogenetic methods. They found that greater seizure power and area triggered SD and the SD, in turn, immediately suppressed the seizure. Based on these findings, they propose a unifying theory wherein SD is a fundamental endogenous antiseizure and protective mechanism in the central nervous system. This unifying theory is easy to understand, but it implies a view that SD may be an integral part of the brain function (not always harmful pathological but sometimes physiological states) and serve fundamental roles (even protective role as antiseizure effect). This point of view is novel and interesting, although further investigation is needed to prove it. The unifying theory and its involved viewpoint may bring new ideas to basic research and clinical work.

There are some confusions and even mistakes in the manuscript that need to be solved.

Major issue :

- 1.The unifying theory should prove to be universal. This study has its limitation. Although wild type and transgenic mice, and three agents used in the research, all results of the study were based on the focal seizure model. To prove it is universal, it would be better to prove that SD can suppress generalized seizure in a generalized seizure animal model induced by systemic agents.
- 2.MK-801 was reported to have antiseizure and anticonvulsant effects before, even in 4-AP induced seizure animal models (Patricia Salazar & Ricardo Tapia. *Neurochemical Research*, 2012, 37:96–603; Keiko Sato Kiyoshi Morimoto Motoi Okamoto, *Brain Research*, 1988, 463:12-20). This research found that MK-801 suppressed SD and therefore enhanced seizure activity. The result is really confusing. What is the reason for the different results?
- 3.Mutations in CaV2.1 voltage-gated calcium channels in familial hemiplegic migraine is known to predispose to seizures. But this study showed that topical application of 4AP in the familial hemiplegic migraine knockin mice induced more SDs than in their wild type littermates, and more importantly, spontaneously developing SDs in FHM1 mutants also suppressed seizure intensity. Their peak ECoG power is lower and the proportion of animals that developed hyperemic transients is also lower. Why do these transgenic mice predispose to seizures?

Minor issue:

- 1.Focal cortical seizures induced by topical application of epileptogenic agents are dependent on the concentration. Then is the rate of animals with SDs triggered by seizures also concentration-dependent?
- 2.The abbreviation of 4-aminopyridine should be unified as 4AP or 4-AP (in the result section page 4).
- 3.The illustrated of Figure 2 D described that there were A total of 11 spontaneous SDs, but Figure 2 C and D showed only 10 spontaneous SDs. Was this one hidden in front of these 10 SDs?
- 4.In Figure 3B, some of the experiments did not last for 240 minutes. Were the animals dead in those experiments?
- 5.Extended data figure 6. A: There was no sign for following words: Averaged ECoG power time courses normalized to pre-SD (5 minutes) seizure power (S) from E1 are shown separately for each frequency band in ChR2+ mice. C: There was no sign of C in Extended data figure 6.
- 6.The article of Reference 6 can not be found online.
- 7.Both wild type littermate and CD1 mice are wild type. Why were the rate of animals induced SDs by 4-AP in WT littermate mice (16.7%, 1/6) much less than CD1 (69%)?

Reviewer #2 (Remarks to the Author):

Summary

This manuscript contributes novel and compelling data on the interplay between seizures and spreading depression (SD), a topic that has been studied since 1953(1) and has increasing relevance today both in the theory of brain function/failure and also in management of patients with acute brain injury. The authors use three different convulsants, topically applied to cortex in mice, to study the occurrence and interplay of seizures and SDs using ECoG recordings and IOS imaging. They find that SDs occur spontaneously, interrupting seizure activity, in a majority of animals, and that SDs are associated with more severe seizures – both in spatial extent (generalization) and amplitude (ECoG power). They show that it is the seizures themselves, and not the convulsants directly, that provoke SDs, in a series of experiments with convulsants and TTX to block the seizures (and thus the SDs). They show that SD, induced remotely either with optogenetics or topical KCl, inhibits ongoing seizure activity, and that even preconditioning with a single SD reduces seizures observed after subsequent application of convulsants. As further evidence, they administer MK-801 to a separate group of mice to suppress spontaneous SDs, and find that this suppression results in greater seizure activity. These data all convincingly support the conclusions that (1) greater acute seizure severity is more likely to cause SD, (2) that SDs are an inherent mechanism involved in seizure expression, and (3) SDs limit seizure activity. The data are presented clearly, illustrations are appropriate, and study methods and results are well-documented. The only major shortcoming of the paper is that conclusions and interpretation reach beyond what is justified by their experimental results.

Major critiques

In this reviewer's opinion, the extensive focus on evolution in this paper is not justified, as it is purely speculative, and should be removed from the title, abstract, and introduction. Any evolutionary implications are speculative at best, and should be reserved for discussion. Further, mention of the "evolutionary pressure that allowed SD to persist", "providing a teleological explanation for the persistence of SD" (Abstract), and "affords a survival advantage" (2nd paragraph Intro) are presumptuous and perhaps misguided. Not all features in biology were positively selected through evolution on the basis of conferring a survival advantage, and thus do not necessarily serve any teleology. Is there a teleologic purpose of seizures, or other diseases or disease mechanisms? Many features in biology are simply vestigial, and/or do not confer sufficient disadvantage to reproduction to be eliminated by natural selection. Any such speculation about evolution should be limited, and placed in the Discussion, with some consideration given to alternatives.

It is unclear why there is a sole focus on migraine, to the exclusion of brain injury, in interpreting relevance of the results. There is no evidence for seizures as the initiating mechanism of migraine aura. It is written that "it is unlikely that all auras are triggered by focal seizures" (Discussion paragraph 3), but in fact it is pure speculation that any auras at all are triggered by seizures. By contrast, there is both clinical and experimental evidence for the interplay of seizures and SDs in both stroke and brain trauma.(2-5) Moreover, in stroke and brain trauma, this interplay continues for hours and even days with multiple repetitive SD events, similar to the prolonged time course of seizures and SDs in these experiments – at least 4 hours (Fig 3B). The aura phase of migraine, by contrast, lasts only 20 min, and only a single SD is thought to occur. Finally, there is prior evidence in brain injury, and not migraine, that seizures are terminated by SDs, e.g. Figure 5.2 in (5), (2) Thus, it would appear that these experimental models are much more relevant to brain injury than to migraine. The introduction and discussion should be edited to better reflect these considerations. "Migraine aura" should be removed from the title since this is not directly addressed, and because the main results are based on a seizure model and not a migraine model. The title should summarize the main experimental findings only.

The narrative throughout the paper is that SDs are beneficial because they suppress seizures. However, this interpretation is not justified. What is the evidence that seizures are more harmful than SDs? To the contrary, there are several lines of evidence that SDs are a more severe pathology than seizures.(5, 6) An alternate interpretation of the data –at least as plausible– is that the onset of SDs reflects a deterioration of tissue health. No data are provided using histology, immunohistochemistry, etc., to suggest that the brain has a better outcome when SDs

suppress seizures than when they do not. For instance, this could be addressed in MK-801 experiments where SDs are suppressed by examining markers of neuronal injury. Without such evidence, the narrative of SD benefit throughout should be edited to reflect a more balanced interpretation confined to data presented and prior literature. [E.g. the last sentence of paragraph 4 in Discussion: (1) we do not know whether SDs or seizures are injurious in this model, as in the Nedergaard paper referenced, (2) how can SDs be considered self-limiting when they recur in high numbers in these models – higher numbers than even some stroke models where they are deleterious.]

It is not clear how the theories presented are “unifying”. They do not take into account stroke and brain injury, nor do they address evidence that SDs can cause seizures.(3, 7)

Methods

After convulsive agents were topically applied, were they then washed out?

Results and Discussion

In the first paragraph, the waveforms of the seizures generated by the three agents should be described, and examples on expanded scales (such as 10 s sample) should be provided in Figure 1. For instance, it is useful to know whether these are true seizures, generally defined as repetitive spiking at frequency $> 3/s$, or in some cases $1/s$. Was it tonic or clonic activity, or a mix? Monomorphic spiking is mentioned, but the frequency is not. Thus, this could be more similar to periodic epileptiform discharges (PEDs) than seizures. The ictal-interictal continuum is complex, entailing a range of abnormal activity, and more detail should be provided to understand what is being referred to as “seizure” in this paper. Burst-suppression is usually not considered seizure: why were they quantified and considered seizure here?

In second paragraph, the duration of ECoG/imaging monitoring should be mentioned. This information is not provided until Figure 3B, and is not mentioned in Methods. Why was monitoring terminated before 4 hr in most animals?

The CBF coupling to SDs should be reported. If there is inverse coupling, it would help explain why suppressions are longer-lasting than in naïve, healthy brain. It would also provide evidence as to whether these SDs may be harmful vs. protective, and how well the experiments model migraine (should be hyperemic) vs acute injury (mixed CBF coupling). Did CBF coupling to SDs change over time, such as in a cluster?

Relatedly, could arrows be added to Figure 4C (top) to show the time of SDs?

Is there any potential explanation for the different DC waveforms of SDs that did not penetrate the SD focus? Is there any prior evidence for such waveforms associated with SD? Was CBF coupling for these events the same as for penetrating SDs that have stereotypical DC shift? Also, it should be explained how the amplitudes of the waveforms were measured, and how it was determined that they were positive or negative (Fig 3F), when waveforms have both positive and negative components. Was it the peak amplitude from baseline that is shown in 3F, and only the larger component (pos or neg) that was measured?

What is the order of the time sequence in Figure 3D? The panels should be labeled with time stamps.

In paragraph 9 of Results, and Figure 7, it is confusing how the spreading of seizures observed after SD induction could be assessed when, in the prior paragraph 8, it is stated that “seizures did not reignite” after the induced SD, and that SD pre-conditioning “prevented the emergence of seizure activity.” In figure 7, the numbers of animals are given, but how many of these animals had seizures (how many contributed data to the graphs)?

Paragraph 12 states that “we could not directly quantify the antiseizure effect of spontaneously occurring SDs because higher seizure power also triggered more SDs, creating a bi-directional relationship.” The reasoning here is not clear. There are two analyses that would be highly informative to the authors’ hypotheses: (1) time sequences of the power changes after the first

spontaneously occurring SD, as done for the induced SDs in Figure 5E (understanding that effects may reflect suppression by multiple SDs), and (2) comparison of overall seizure burden (measured as time series of power throughout experiment) between animals with spontaneous SDs and those without (n=22 vs. n=10 in 4AP group). This would illustrate antiseizure effects of SDs as well as the propensity for mice with higher delta power (more seizures) to generate SDs.

Sentence 3, paragraph 5 of Discussion misses the main result of the paper that SDs are the more common finding (50% of patients vs. 17% with seizures) and that seizures occur almost exclusively in patients with SDs. This is evidence that SDs usually occur without seizures, but that they may facilitate their occurrence.

"Is SD Physiological?" The points of this section and this question are not clear. Are the authors suggesting that SD may play a role in normal physiologic function? The examples cited are all related to pathology (e.g. induced by epileptic spikes, in genetically susceptible mice), not normal function. Or do the authors mean to suggest that not all SDs have adverse effects on brain tissue health? If so, this point is already widely recognized.(8-11)

1. Van Harreveld A, Stamm JS. Spreading cortical convulsions and depressions. *J Neurophysiol.* 1953;16(4):352-66.
2. Fabricius M, Fuhr S, Willumsen L, Dreier JP, Bhatia R, Boutelle MG, et al. Association of seizures with cortical spreading depression and peri-infarct depolarisations in the acutely injured human brain. *Clin Neurophysiol.* 2008;119(9):1973-84.
3. Dreier JP, Major S, Pannek HW, Woitzik J, Scheel M, Wiesenthal D, et al. Spreading convulsions, spreading depolarization and epileptogenesis in human cerebral cortex. *Brain.* 2012;135(Pt 1):259-75.
4. Hartings JA, Bullock MR, Okonkwo DO, Murray LS, Murray GD, Fabricius M, et al. Spreading depolarisations and outcome after traumatic brain injury: a prospective observational study. *Lancet Neurol.* 2011;10(12):1058-64.
5. Revankar GS, Winkler MKL, Major S, Schoknecht K, Heinemann U, Woitzik J, et al. Spreading depolarizations and seizures in clinical subdural electrocorticographic recordings. *Seizures in Critical Care* 2016.
6. Dreier JP, Isele T, Reiffurth C, Offenhauser N, Kirov SA, Dahlem MA, et al. Is spreading depolarization characterized by an abrupt, massive release of Gibbs free energy from the human brain cortex? *Neuroscientist.* 2013;19(1):25-42.
7. Gorji A, Speckmann EJ. Spreading depression enhances the spontaneous epileptiform activity in human neocortical tissues. *Eur J Neurosci.* 2004;19(12):3371-4.
8. Shuttleworth CW, Andrew RD, Akbari Y, Ayata C, Balu R, Brennan KC, et al. Which Spreading Depolarizations Are Deleterious To Brain Tissue? *Neurocrit Care.* 2020;32(1):317-22.
9. Dreier JP, Reiffurth C. The Stroke-Migraine Depolarization Continuum. *Neuron.* 2015;86(4):902-22.
10. Dreier JP. The role of spreading depression, spreading depolarization and spreading ischemia in neurological disease. *Nat Med.* 2011;17(4):439-47.
11. Nedergaard M, Hansen AJ. Spreading depression is not associated with neuronal injury in the normal brain. *Brain Res.* 1988;449(1-2):395-8.

We thank the Referees again for their constructive critiques and greatly appreciate the opportunity to
respond and revise the manuscript accordingly. We took each suggestion to heart and addressed them as
outlined point-by-point below. All revisions are marked by red font in the text.

**Reviewer #1:**

*Spreading depression (SD) occurs in neurological disorders such as migraine, Epilepsy and brain injury. Is
it just a concomitant phenomenon, or harmful as many studies believe, or protective as this study
suggests? In this study, authors used in vivo mouse model of focal seizures induced by 4-AP, PG and BIC
both in wild type and transgenic mice, and electrophysiological, IOS imaging and optogenetic methods.
They found that greater seizure power and area triggered SD and the SD, in turn, immediately suppressed
the seizure. Based on these findings, they propose a unifying theory wherein SD is a fundamental
endogenous antiseizure and protective mechanism in the central nervous system. This unifying theory is
easy to understand, but it implies a view that SD may be an integral part of the brain function (not always
harmful pathological but sometimes physiological states) and serve fundamental roles (even protective
role as antiseizure effect). This point of view is novel and interesting, although further investigation is
needed to prove it. The unifying theory and its involved viewpoint may bring new ideas to basic research
and clinical work.*

*There are some confusions and even mistakes in the manuscript that need to be solved.*

**Major issue :**

*1.The unifying theory should prove to be universal. This study has its limitation. Although wild type and
transgenic mice, and three agents used in the research, all results of the study were based on the focal
seizure model. To prove it is universal, it would be better to prove that SD can suppress generalized seizure
in a generalized seizure animal model induced by systemic agents.*

Referee's point about generalized seizures is well taken. We have provided a discussion of previous work
showing that SDs do occur in generalized seizure models and in turn suppress the seizures (page 9,
*Evidence from other model systems*). However, prompted by the Referee's suggestion, we performed new
experiments to directly test the theory in a generalized seizure model. We used systemic kainate injection
as a ubiquitous model to induce generalized seizures and performed electrophysiological recordings from
hippocampal CA1 region.

All mice developed generalized seizures detected in hippocampal CA1 after systemic kainate. Remarkably,
seizures quickly culminated in hippocampal SD in 100% of the animals. More importantly, SD occurrence
suppressed hippocampal seizures for at least 30 minutes despite continued systemic presence of a highly
epileptogenic drug. We now present these data in the manuscript (page 8, paragraph 1, Figure 10). We
thank the Referee for this excellent suggestion and hope the data will help bolster our theory.

*2.MK-801 was reported to have antiseizure and anticonvulsant effects before, even in 4-AP induced seizure
animal models (Patricia Salazar & Ricardo Tapia. *Neurochemical Research*, 2012, 37:96–603; Keiko Sato
Kiyoshi Morimoto Motoi Okamoto, *Brain Research*, 1988, 463:12-20). This research found that MK-801
suppressed SD and therefore enhanced seizure activity. The result is really confusing. What is the reason
for the different results?*

Another excellent point. We performed this experiment to seek proof-of-principle that when the tissue is
incapable of developing SD, seizures would worsen. For this, we needed a tool to block SD occurrence

without suppressing seizures. When we decided to test MK-801 (1-3 mg/kg), a potent SD inhibitor, we did
not know whether it would also suppress seizures in our model, because, in the literature, the antiseizure
effect of MK-801 has been model-dependent.

For example, Salazar and Tapia¹ (and studies from Tapia et al. in general) used microdialysis of 4AP in the
rat hippocampus. Hippocampus has a high density of NMDA receptors and their contribution to seizure
induction might have differed from the parietooccipital cortex we studied. Their procedure was also more
invasive compared with our topical 4AP application, and could have triggered one or more SDs during the
probe placement that may have confounded seizure progression and response to MK-801. Sato et al.²
used a kindling model, rather than 4AP, in rats. Kindling, like long-term potentiation, is likely dependent
on NMDA receptors. Indeed, they concluded in their Discussion that "... the action of MK-801 was not
elevating the threshold of seizure generalization but served a direct role in the initiation of epileptic
activity." Consistent with this, in our study MK-801 enhanced seizure activity only at the remote sites
(Figure 9B, E2 and E3). At the seizure focus, there was a mixed, frequency band-dependent effect (Figure
9B, E1, 0-8 Hz band suppressed), suggesting that inhibition of SD by MK-801 enhanced seizure
generalization rather than generation.

In contrast to these studies, many others found MK-801 ineffective. For example, MK-801 did not suppress
seizures after systemic bicuculline³ or 4AP^{4,5,6} administration, and even augmented pilocarpine induced
limbic seizures,^{7,8} kainate seizures,^{9,10} and absence seizures in genetic models.¹¹

In summary, MK-801 may indeed have antiseizure effects in other models and systems but did not do so
consistently in our model, at least at the doses tested here in the Swiss outbred CD1 mouse strain, which
requires higher doses than other strains to suppress seizures.¹² We used it as a tool to test a specific
hypothesis and it worked for us.

*3. Mutations in CaV2.1 voltage-gated calcium channels in familial hemiplegic migraine is known to*
*predispose to seizures. But this study showed that topical application of 4AP in the familial hemiplegic*
*migraine knockin mice induced more SDs than in their wild type littermates, and more importantly,*
*spontaneously developing SDs in FHM1 mutants also suppressed seizure intensity. Their peak ECoG power*
*is lower and the proportion of animals that developed hyperemic transients is also lower. Why do these*
*transgenic mice predispose to seizures?*

FHM1 mutations do predispose to seizures, presumably because of increased glutamate release due to
gain-of-function mutations in the Ca_v2.1 channel. But FHM1 mice are also more susceptible to SD. So, it
was not surprising to see more SDs in FHM1 mice in the 4AP model. The Referee is correct in that both
the peak EEG power and hyperemic transients were lower in FHM1 mice. We believe this was because
early and more frequent spontaneous SDs kept both in check.

Awake FHM mice develop ostensibly spontaneous SDs.^{13,14} Our data suggest that seizures may be the
inciting event triggering these spontaneous SDs. As such, when a focal seizure culminates in SD, it quells
the seizure and/or its generalization. When this happens, we see an SD rather than the overt signs of
generalized seizures. Of course, when seizure susceptibility is high, even the increased SD susceptibility is
unlikely to keep it in check completely and breakthrough seizures might occur.

*Minor issue:*
*1. Focal cortical seizures induced by topical application of epileptogenic agents are dependent on the*
*concentration. Then is the rate of animals with SDs triggered by seizures also concentration-dependent?*

In addition to 100 mM 4AP and 5 mM BIC, we tested 30 mM (n=4) and 5 mM (n=2) 4AP, and 1 mM (n=1),
0.5 mM (n=1) and 5 μ M (n=1) BIC. None of these lower concentrations triggered an SD, suggesting that
SD occurrence was concentration dependent. This is now briefly stated in the Results section (page 4,
paragraph 2). However, we did not test whether even higher concentrations can trigger more frequent
SDs.

*2.The abbreviation of 4-aminopyridine should be unified as 4AP or 4-AP (in the result section page 4).*

Done throughout, thank you.

*3.The illustrated of Figure 2 D described that there were A total of 11 spontaneous SDs, but Figure 2 C and*
*D showed only 10 spontaneous SDs. Was this one hidden in front of these 10 SDs?*

One of the vertical dashed lines marking each SD on Figure 2 (left panel) was missing (third SD). This is
now fixed, making it a total of 11. We apologize for this mistake.

*4.In Figure 3B, some of the experiments did not last for 240 minutes. Were the animals dead in those*
*experiments?*

Maximum duration of recordings was 240 minutes. We stopped the recordings if epileptiform activity
diminished (<3 spikes/20 sec) for at least 10 minutes suggesting waning of drug effect (n=10). Three
experiments were shorter (76, 90 and 117 minutes) because these were additional controls done at later
stages of the project to ensure stability of the model system, and to record high quality movies. Only one
wild type and two FHM1 mice died unexpectedly during the recordings. These are now detailed in the
Methods (pages 12-13).

*5.Extended data figure 6. A: There was no sign for following words: Averaged ECoG power time courses*
*normalized to pre-SD (5 minutes) seizure power (S) from E1 are shown separately for each frequency band*
*in Chr2+ mice. C: There was no sign of C in Extended data figure 6.*

We apologize for these mistakes. We removed the label “S” from the legend as this was spelled out on
each figure as “Seizure”. We added the panel label “C”.

*6.The article of Reference 6 can not be found online.*

We once again apologize for this. The bioRxiv paper was revised and can be found at:
<https://doi.org/10.1101/455519> (<https://www.biorxiv.org/content/10.1101/455519v2>). This is also
corrected in the Bibliography.

*7.Both wild type littermate and CD1 mice are wild type. Why were the rate of animals induced SDs by 4-*
*AP in WT littermate mice (16.7%, 1/6) much less than CD1 (69%)?*

We are not sure, but the difference is likely related to the wild type genetic background (CD1 vs. C57B6),
or older age in the WT littermates (8.3 \pm 1.6 months) compared with CD1 (3.2 \pm 0.1 months), which is known
to elevate SD threshold and decrease SD susceptibility.^{15, 16, 17}

**Reviewer #2:**

*Summary*

*This manuscript contributes novel and compelling data on the interplay between seizures and spreading*
*depression (SD), a topic that has been studied since 1953(1) and has increasing relevance today both in*
*the theory of brain function/failure and also in management of patients with acute brain injury. The*
*authors use three different convulsants, topically applied to cortex in mice, to study the occurrence and*
*interplay of seizures and SDs using ECoG recordings and IOS imaging. They find that SDs occur*
*spontaneously, interrupting seizure activity, in a majority of animals, and that SDs are associated with*
*more severe seizures – both in spatial extent (generalization) and amplitude (ECoG power). They show that*
*it is the seizures themselves, and not the convulsants directly, that provoke SDs, in a series of experiments*
*with convulsants and TTX to block the seizures (and thus the SDs). They show that SD, induced remotely*
*either with optogenetics or topical KCl, inhibits ongoing seizure activity, and that even preconditioning*
*with a single SD reduces seizures observed after subsequent application of convulsants. As further*
*evidence, they administer MK-801 to a separate group of mice to suppress spontaneous SDs, and find that*
*this suppression results in greater seizure activity. These data all convincingly support the conclusions that*
*(1) greater acute seizure severity is more likely to cause SD, (2) that SDs are an inherent mechanism*
*involved in seizure expression, and (3) SDs limit seizure activity. The data are presented clearly, illustrations*
*are appropriate, and study methods and results are well-documented. The only major shortcoming of the*
*paper is that conclusions and interpretation reach beyond what is justified by their experimental results.*

*Major critiques*

*In this reviewer’s opinion, the extensive focus on evolution in this paper is not justified, as it is purely*
*speculative, and should be removed from the title, abstract, and introduction. Any evolutionary*
*implications are speculative at best, and should be reserved for discussion.*

*Further, mention of the “evolutionary pressure that allowed SD to persist”, “providing a teleological*
*explanation for the persistence of SD” (Abstract), and “affords a survival advantage” (2nd paragraph Intro)*
*are presumptuous and perhaps misguided. Not all features in biology were positively selected through*
*evolution on the basis of conferring a survival advantage, and thus do not necessarily serve any teleology.*
*Is there a teleologic purpose of seizures, or other diseases or disease mechanisms? Many features in*
*biology are simply vestigial, and/or do not confer sufficient disadvantage to reproduction to be eliminated*
*by natural selection. Any such speculation about evolution should be limited, and placed in the Discussion,*
*with some consideration given to alternatives.*

We revised the manuscript by limiting the topic of evolutionary pressure to Discussion.

*It is unclear why there is a sole focus on migraine, to the exclusion of brain injury, in interpreting relevance*
*of the results. There is no evidence for seizures as the initiating mechanism of migraine aura. It is written*
*that “it is unlikely that all auras are triggered by focal seizures” (Discussion paragraph 3), but in fact it is*
*pure speculation that any auras at all are triggered by seizures. By contrast, there is both clinical and*
*experimental evidence for the interplay of seizures and SDs in both stroke and brain trauma.(2-5)*
*Moreover, in stroke and brain trauma, this interplay continues for hours and even days with multiple*
*repetitive SD events, similar to the prolonged time course of seizures and SDs in these experiments – at*
*least 4 hours (Fig 3B). The aura phase of migraine, by contrast, lasts only 20 min, and only a single SD is*
*thought to occur. Finally, there is prior evidence in brain injury, and not migraine, that seizures are*
*terminated by SDs, e.g. Figure 5.2 in (5), (2) Thus, it would appear that these experimental models are*
*much more relevant to brain injury than to migraine. The introduction and discussion should be edited to*
*better reflect these considerations. “Migraine aura” should be removed from the title since this is not*

*directly addressed, and because the main results are based on a seizure model and not a migraine model.*
*The title should summarize the main experimental findings only.*

These are great points that we will try to clarify individually:

- 1) We beg to disagree that migraine aura was “... *not directly addressed, ...*”. Our study is relevant to
migraine aura to the extent that SD is the mechanism underlying aura. SD is the most commonly
used migraine model, so our study did use a migraine model as well as a seizure model. There is
ample data indicating that migraine is a disorder of hyperexcitability, hence the similitude to
seizures.^{18, 19, 20, 21, 22, 23, 24, 25, 26, 27, 28, 29, 30, 31}
- 2) Our “*sole focus on migraine, to the exclusion of brain injury*” was because our model has no injury
component or vascular/metabolic compromise. If we focused on brain injury, we would have
been criticized for our model not representing brain injury.
- 3) It is true that “*There is no evidence for seizures as the initiating mechanism of migraine aura.*”
Hence, our study puts forth a novel theory based on data. The “... *speculation that any auras at*
*all are triggered by seizures.*” was based on the data presented in the manuscript. As such, it was
not a speculation but rather an interpretation and clinical extrapolation of experimental data.
- 4) It is true that “*The aura phase of migraine, by contrast, lasts only 20 min, and only a single SD is*
*thought to occur.*” But the pharmacological seizure model we employed creates a severe focal
epileptogenic environment that lasts for hours. It is likely that epileptic events in otherwise non-
injured brains could trigger a single SD and be suppressed by it, leading to a single event perceived
as migraine aura.
- 5) It is also true that “... *there is prior evidence in brain injury, and not migraine, that seizures are*
*terminated by SDs, ...*”. But how could there be such evidence in migraine when invasive
recordings are not possible? Absence of evidence for something is not evidence for absence of
that thing.

Nevertheless, in the revised manuscript we removed migraine aura from the title and briefly stated the
implications of our data for brain injury in Discussion (pages 10-11).

*The narrative throughout the paper is that SDs are beneficial because they suppress seizures. However,*
*this interpretation is not justified. What is the evidence that seizures are more harmful than SDs? To the*
*contrary, there are several lines of evidence that SDs are a more severe pathology than seizures.(5, 6)*

We did not claim that SDs are always beneficial, but rather suppression of seizures and their generalization
by SD can be beneficial. Benefit, at the level of the whole organism, is not only measured by the degree
of tissue injury (i.e., excitotoxic cell death). Seizures, especially if they become generalized, can be more
harmful (and even terminal) for an organism than a migraine aura because of loss of consciousness,
convulsions leading to bodily harm, and prolonged post-ictal encephalopathy etc. Therefore, the
interpretation that SD can be beneficial in the context of seizures we believe is justified.

We now clarify these points in the second paragraph of the Discussion (page 8).

Besides, seizures, when prolonged, are well known to cause excitotoxic cell death. No such evidence exists
for SD, unless the tissue has already suffered some form of injury and/or metabolically compromised. In
migraine, with no coincident injury or metabolic compromise, as is the case in our experimental model,
aura is not thought of as a damaging event (perhaps with the rare exception of migrainous infarction,
which is a diagnosis of exclusion). The citation by the Referee³² elegantly outlines the burden of SD on
cells. But if that was sufficient to be injurious by itself, then one would expect migraine aura attacks to

cause brain damage. This clearly is not the case. Recurrent SDs may be observed in the setting of severe
pathology, but this does not automatically mean they cause pathology by themselves.

*An alternate interpretation of the data –at least as plausible– is that the onset of SDs reflects a*
*deterioration of tissue health.*

This is certainly possible. But we believe it is unlikely because we observed an SD in some cases within
mere minutes after seizure onset, at a time when the tissue was robust and healthy. The same tissue in
most cases was capable of sustaining seizures thereafter. Nevertheless, we now mention this possibility
in the Discussion (page 10, *Potential mechanisms*). What we are certain is that cellular ionic gradients
inexorably begin to collapse during seizures, and such partial depolarization can lead to SD. The process
can be fully reversible without injury, although the energy demand during a period of energy debt can
delay recovery.

*No data are provided using histology, immunohistochemistry, etc., to suggest that the brain has a better*
*outcome when SDs suppress seizures than when they do not. For instance, this could be addressed in MK-*
*801 experiments where SDs are suppressed by examining markers of neuronal injury. Without such*
*evidence, the narrative of SD benefit throughout should be edited to reflect a more balanced interpretation*
*confined to data presented and prior literature. [E.g. the last sentence of paragraph 4 in Discussion: (1) we*
*do not know whether SDs or seizures are injurious in this model, as in the Nedergaard paper referenced, ...*

Benefit, at the level of the whole organism, is not only measured by the degree of direct tissue injury (i.e.,
excitotoxic cell death). Seizures, especially if they become generalized, can be more harmful (and even
terminal) for an organism than a migraine aura because of loss of consciousness, convulsions leading to
bodily harm, and prolonged post-ictal encephalopathy etc. In other words, prevention of seizure
generalization has benefits independent of direct cell death. Therefore, we do not think demonstration
of reduced excitotoxic cell death/injury is a prerequisite to implicate a beneficial effect in this context.

Whether SD indeed exacerbates or ameliorates seizure-induced cell death is an interesting question in
itself, although MK-801 is probably not the best way to test this hypothesis because it may have direct
protective effects on seizure-induced tissue injury as well.

*... (2) how can SDs be considered self-limiting when they recur in high numbers in these models – higher*
*numbers than even some stroke models where they are deleterious.]*

We used the term self-limiting for SD in the sense that they last <1 minute (in the absence of coincident
injury or metabolic compromise). SDs occurred in high numbers herein but this is a severe
pharmacological seizure model that is unlikely to occur in migraineurs. Besides, there is no metabolic
compromise in our model as there is in stroke.

*It is not clear how the theories presented are “unifying”. They do not take into account stroke and brain*
*injury, nor do they address evidence that SDs can cause seizures.(3, 7)*

A unifying theory does not have to unify everything. We used the term “unifying” in the context of
migraine aura (SD) – seizure continuum. Our theory provides a potential trigger for aura in migraineurs as
well as a potential teleological explanation for persistence of SD. It lays out a bidirectional feedback
relationship between SD and seizures in non-injured brain. This is what we referred to as “unifying.”

It is true that in injured brain both SDs and seizures occur interchangeably. However, in injured brain,
when an SD emerges during a seizure, we cannot really know whether the emerging SD was indeed
triggered by the preceding seizure or by the underlying injury itself. Similarly, when a seizure emerges
after an SD, we cannot really know whether the seizure was triggered by the preceding SD or by the
underlying injury. In fact, we are not aware of seizure activity triggered by an SD in an otherwise normal
brain (except for occasional transient runs of afterdischarges). This is why we focused on migraine rather
than injured brain. In the latter, there is a potential trigger for both seizures and SD independently,
confounding the direct interactions between the two.

We nevertheless appreciate the Reviewer's point and have revised the manuscript to introduce the
unifying theory only in Discussion.

*Methods*

*After convulsive agents were topically applied, were they then washed out?*

No. In all experiments, topical application was performed twice ten minutes apart without any subsequent
washing out. This is now clarified in the Methods (page 12, *Drugs*).

*Results and Discussion*

*In the first paragraph, the waveforms of the seizures generated by the three agents should be described,*
*and examples on expanded scales (such as 10 s sample) should be provided in Figure 1. For instance, it is*
*useful to know whether these are true seizures, generally defined as repetitive spiking at frequency > 3/s,*
*or in some cases 1/s. Was it tonic or clonic activity, or a mix? Monomorphic spiking is mentioned, but the*
*frequency is not. Thus, this could be more similar to periodic epileptiform discharges (PEDs) than seizures.*
*The ictal-interictal continuum is complex, entailing a range of abnormal activity, and more detail should*
*be provided to understand what is being referred to as "seizure" in this paper. Burst-suppression is usually*
*not considered seizure: why were they quantified and considered seizure here?*

We thank the Reviewer for giving us the opportunity to clarify.

First, please note that our highly focal recording of cortical electrical activity using a glass micropipette
(~10 μm tip) placed on cortical surface is fundamentally different from classical scalp and even subdural
EEG using orders-of-magnitude larger disc electrodes where such definitions of epileptiform activity and
ictal-interictal continuum are based on. It is not possible to apply traditional EEG criteria because the
morphology of electrophysiological activity is different. For the same reason, we realized that our use of
the term "burst-suppression" was probably not justified. What we meant was periodic bursts, which is
how we now refer to them in the revised manuscript.

With these in mind, the local field potentials in the convulsant exposed area showed rhythmic activity that
was consistent with electrographic seizure. After 4AP, the electrographic pattern consisted of large
amplitude (~300% over baseline) rhythmic discharges that started at ~1 Hz and evolved to ~4 Hz over a
period of ~20 seconds before attenuation. The cycle repeated itself over a ~30-minute period. After PG,
the electrographic pattern started out as ~0.5 Hz spike wave activity with an amplitude ~250% over
baseline and evolved in amplitude to ~600% over baseline with continuous ~0.5 Hz activity. This pattern
developed over a ~70-minute period. After BIC, the electrographic pattern started as ~0.2 Hz spike-wave
activity with an amplitude of ~20% over baseline and evolved to ~1-2 Hz spike-wave activity with an
amplitude of ~250%. This pattern developed over a ~40-minute period.

Although these events may not fully conform to the criteria for electrographic seizures recorded by scalp
electrodes, in the context of micropipette field potential recordings, they demonstrate evolving
rhythmicity that is consistent with synchronized network excitability and seizure. Moreover, such
electrophysiological activity induced by focal 4-AP, PG and BIC application is widely referred to as seizures
in the literature.^{33, 34, 35, 36}

These are now detailed in the Results (page 4, paragraph 1). Because Figure 1 was already very large and
busy, we now provide representative electrophysiological tracings on expanded scales as a new
supplemental figure (Supplemental Figure 2).

*In second paragraph, the duration of ECoG/imaging monitoring should be mentioned. This information is*
*not provided until Figure 3B, and is not mentioned in Methods. Why was monitoring terminated before 4*
*hr in most animals?*

Maximum duration of recordings was 240 minutes. We stopped the recordings if epileptiform activity
diminished (<3 spikes/20 sec) for at least 10 minutes suggesting waning of drug effect (n=10). Three
experiments were shorter (76, 90 and 117 minutes) because these were additional controls done at later
stages of the project to ensure stability of the model system, and to record high quality movies. Only one
wild type and two FHM1 mice died unexpectedly during the recordings. These are now stated in the
Results (page 3, last line) and Methods (pages 12-13).

*The CBF coupling to SDs should be reported. If there is inverse coupling, it would help explain why*
*suppressions are longer-lasting than in naïve, healthy brain. It would also provide evidence as to whether*
*these SDs may be harmful vs. protective, and how well the experiments model migraine (should be*
*hyperemic) vs acute injury (mixed CBF coupling). Did CBF coupling to SDs change over time, such as in a*
*cluster?*

Unfortunately, we did not record the CBF. However, intrinsic optical signal imaging, reflecting total Hb
concentration, and therefore, cortical blood volume (CBV), showed a triphasic response to the first SD
(initial hypoperfusion, followed by transient normalization, and a longer lasting hypoperfusion) and a
predominantly hyperemic response to all subsequent SDs without a particular evolution to inverse
coupling in case of clusters. This is the typical response in healthy mouse cortex,³⁷ suggesting that these
SDs were not harmful and that the model approximates migraine rather than brain injury. We now present
a representative experiment in the new Supplemental Figure 6 and state the findings in the Results (page
6, paragraph 1).

*Relatedly, could arrows be added to Figure 4C (top) to show the time of SDs?*

In Figure 4, we showed an experiment in which no SD occurred in order to demonstrate hyperemic
transients during seizure activity without the confounding effect of an SD. The new Supplemental Figure
6 now shows an experiment with SDs, where arrows are added.

*Is there any potential explanation for the different DC waveforms of SDs that did not penetrate the SD*
*focus? Is there any prior evidence for such waveforms associated with SD?*

We do not know the exact mechanism creating positive DC potential shifts when SDs did not penetrate
the seizure focus (based on intrinsic optical signal imaging); these SDs were still associated with the typical

negative DC shifts at the anterior recording site. We might nevertheless speculate that it is the dipole of
the tremendous sink created by the SD approaching and surrounding the focus.

*Was CBF coupling for these events the same as for penetrating SDs that have stereotypical DC shift?*

Please see our response above regarding “CBF coupling to SDs.” When an SD did not penetrate the seizure
focus (i.e., drug application site), it did not trigger any hemodynamic change in the focus. Outside of the
focus, these same SDs evoked typical hemodynamic changes for healthy mouse cortex. No evidence of
inverse coupling was observed. We now show this in the new Supplemental Figure 6 and state the findings
in the Results (page 6, paragraph 1).

*Also, it should be explained how the amplitudes of the waveforms were measured, and how it was*
*determined that they were positive or negative (Fig 3F), when waveforms have both positive and negative*
*components. Was it the peak amplitude from baseline that is shown in 3F, and only the larger component*
*(pos or neg) that was measured?*

We have measured the DC waveforms peak-to-peak and marked as positive or negative based on the
predominant component. Prompted by this question, however, we revisited and measured the
amplitudes from baseline to the predominant peak (positive or negative). We revised Figure 3 and
Supplemental Figure 9 to reflect the new measurements. We thank the Reviewer for bringing up this
question.

*What is the order of the time sequence in Figure 3D? The panels should be labeled with time stamps.*

Done, thank you.

*In paragraph 9 of Results, and Figure 7, it is confusing how the spreading of seizures observed after SD*
*induction could be assessed when, in the prior paragraph 8, it is stated that “seizures did not reignite” after*
*the induced SD, and that SD pre-conditioning “prevented the emergence of seizure activity. In figure 7, the*
*numbers of animals are given, but how many of these animals had seizures (how many contributed data*
*to the graphs)?”*

We apologize for the confusion. We have said “In most cases, seizures did not reignite ...” Some animals
did show seizure activity but at a depressed level, as evident in some panels of Figure 6. All animals shown
in Figures 5 and 6 contributed to Figure 7 (except 1 animal in PG group with induced SD at -5 minutes, due
to a technical issue with IOS imaging), hence our ability to assess spread. As expected from a depressed
level of seizure activity at the drug application site, involvement of remote sites was also reduced. We
now tried to clarify this point in the text (page 6, paragraphs 3-4).

*Paragraph 12 states that “we could not directly quantify the antiseizure effect of spontaneously occurring*
*SDs because higher seizure power also triggered more SDs, creating a bi-directional relationship.” The*
*reasoning here is not clear. There are two analyses that would be highly informative to the authors’*
*hypotheses: (1) time sequences of the power changes after the first spontaneously occurring SD, as done*
*for the induced SDs in Figure 5E (understanding that effects may reflect suppression by multiple SDs), and*
*(2) comparison of overall seizure burden (measured as time series of power throughout experiment)*
*between animals with spontaneous SDs and those without (n=22 vs. n=10 in 4AP group). This would*
*illustrate antiseizure effects of SDs as well as the propensity for mice with higher delta power (more*
*seizures) to generate SDs.*

Prompted by this query, we returned to our data and examined seizure power after the first spontaneous
SD and found a significant suppression compared with animals that did not develop an SD after 4AP. These
data are now presented in the new Figure 8. We thank the Referee for this suggestion.

*Sentence 3, paragraph 5 of Discussion misses the main result of the paper that SDs are the more common*
*finding (50% of patients vs. 17% with seizures) and that seizures occur almost exclusively in patients with*
*SDs. This is evidence that SDs usually occur without seizures, but that they may facilitate their occurrence.*

It is true that SD was the more commonly detected finding than seizures at the recording site(s). But SDs
propagate, while focal seizures don't. It is more likely for an electrode to detect SDs because they
propagate to the recording site(s). Focal seizures at remote locations not covered by the electrode(s) can
be easily missed. Hence, detecting an SD but not a seizure in a patient does not mean there was no focal
seizure activity where the SD originated – it just wasn't picked up by the electrode(s). Moreover, clinical
recordings are obtained by rather large electrodes with contacts that are millimeters in diameter; they
effectively summate the field potentials and can miss microseizures. Therefore, "SDs are the more
common finding" does not necessarily mean that SDs usually occur without seizures.

*"Is SD Physiological?" The points of this section and this question are not clear. Are the authors suggesting*
*that SD may play a role in normal physiologic function? The examples cited are all related to pathology*
*(e.g. induced by epileptic spikes, in genetically susceptible mice), not normal function. Or do the authors*
*mean to suggest that not all SDs have adverse effects on brain tissue health? If so, this point is already*
*widely recognized.(8-11)*

We apologize for the ambiguous subtitle. We did mean to pose the question whether SD could be serving
a function in brain (*physiological*: characteristic of or appropriate to an organism's healthy or normal
functioning; Merriam-Webster 2020). Any "normal" tissue maintains homeostasis with the help of certain
processes keeping others in check. For example, excitatory neurotransmission is physiological to the
extent inhibitory neurotransmission keeps it in check. Both are physiological, but too much of either is
deemed pathological. In the context of our study, we posit that SD may be the process that keeps seizure
activity in check, extinguishing it when it reaches a certain intensity threshold, preventing it from
becoming more generalized. There are many physiological (i.e., beneficial) processes that can be
pathological when overactive or underactive or when they occur at the wrong place and time (e.g.,
inflammation), and SD might be one of those. When it occurs in the wrong context (e.g., injured brain), it
may have pathological consequences. We now tried to clarify this in Discussion (pages 10-11).

REFERENCES

1. Salazar P, Tapia R. Allopregnanolone potentiates the glutamate-mediated seizures induced by 4-
aminopyridine in rat hippocampus in vivo. *Neurochem Res* **37**, 596-603 (2012).

2. Sato K, Morimoto K, Okamoto M. Anticonvulsant action of a non-competitive antagonist of NMDA
receptors (MK-801) in the kindling model of epilepsy. *Brain Res* **463**, 12-20 (1988).
- 3. Rejdak K, Rejdak R, Kleinrok Z, Sieklucka-Dziuba M. The influence of MK-801 on bicuculline evoked
seizures in adult mice exposed to transient episode of brain ischemia. *J Neural Transm (Vienna)*
**107**, 947-957 (2000).
- 4. Juhng KN, *et al.* Induction of seizures by the potent K⁺ channel-blocking scorpion venom peptide
toxins tityustoxin-K(alpha) and pandinustoxin-K(alpha). *Epilepsy Res* **34**, 177-186 (1999).
- 5. Yamaguchi S, Rogawski MA. Effects of anticonvulsant drugs on 4-aminopyridine-induced seizures
in mice. *Epilepsy Res* **11**, 9-16 (1992).
- 6. Wardley-Smith B, Wann KT. Effects of four drugs on 4-aminopyridine seizures: a comparison with
their effects on HPNS. *Undersea biomedical research* **18**, 413-419 (1991).
- 7. Hughes P, Young D, Dragunow M. MK-801 sensitizes rats to pilocarpine induced limbic seizures
and status epilepticus. *Neuroreport* **4**, 314-316 (1993).
- 8. Lee MG, Chou JY, Lee KH, Choi BJ, Kim SK, Kim CY. MK-801 augments pilocarpine-induced
electrographic seizure but protects against brain damage in rats. *Prog Neuropsychopharmacol Biol*
*Psychiatry* **21**, 331-344 (1997).
- 9. Stafstrom CE, Tandon P, Hori A, Liu Z, Mikati MA, Holmes GL. Acute effects of MK801 on kainic
acid-induced seizures in neonatal rats. *Epilepsy Res* **26**, 335-344 (1997).
- 10. Fariello RG, Golden GT, Smith GG, Reyes PF. Potentiation of kainic acid epileptogenicity and
sparing from neuronal damage by an NMDA receptor antagonist. *Epilepsy Res* **3**, 206-213 (1989).
- 11. Maheshwari A, Marks RL, Yu KM, Noebels JL. Shift in interictal relative gamma power as a novel
biomarker for drug response in two mouse models of absence epilepsy. *Epilepsia* **57**, 79-88 (2016).
- 12. Deutsch SI, Mastropaolo J, Powell DG, Rosse RB, Bachus SE. Inbred Mouse Strains Differ in Their
Sensitivity to an Antiseizure Effect of MK-801. *Clinical neuropharmacology* **21**, 255-257 (1998).
- 13. Loonen ICM, *et al.* Brainstem spreading depolarization and cortical dynamics during fatal seizures
in Cacna1a S218L mice. *Brain* **142**, 412-425 (2019).
- 14. Jansen NA, Dehghani A, Linssen MML, Breukel C, Tolner EA, van den Maagdenberg A. First FHM3
mouse model shows spontaneous cortical spreading depolarizations. *Annals of clinical and*
*translational neurology* **7**, 132-138 (2020).
- 15. Clark D, *et al.* Impact of aging on spreading depolarizations induced by focal brain ischemia in rats.
*Neurobiology of aging* **35**, 2803-2811 (2014).
- 16. Guedes RC, Amorim LF, Teodosio NR. Effect of aging on cortical spreading depression. *Braz J Med*
*Biol Res* **29**, 1407-1412 (1996).

17. Hertelendy P, *et al.* Advancing age and ischemia elevate the electric threshold to elicit spreading
depolarization in the cerebral cortex of young adult rats. *J Cereb Blood Flow Metab* **37**, 1763-1775
(2017).
- 18. Siniatchkin M, Averkina N, Andrasik F, Stephani U, Gerber WD. Neurophysiological reactivity
before a migraine attack. *Neurosci Lett* **400**, 121-124 (2006).
- 19. Schoenen J, Wang W, Albert A, Delwaide PJ. Potentiation instead of habituation characterizes
visual evoked potentials in migraine patients between attacks. *European Journal of Neurology* **2**,
115-122 (1995).
- 20. Wang W, Timsit-Berthier M, Schoenen J. Intensity dependence of auditory evoked potentials is
pronounced in migraine: an indication of cortical potentiation and low serotonergic
neurotransmission? *Neurology* **46**, 1404-1409 (1996).
- 21. Ambrosini A, Rossi P, De Pasqua V, Pierelli F, Schoenen J. Lack of habituation causes high intensity
dependence of auditory evoked cortical potentials in migraine. *Brain* **126**, 2009-2015 (2003).
- 22. Ozkul Y, Uckardes A. Median nerve somatosensory evoked potentials in migraine. *Eur J Neurol* **9**,
227-232 (2002).
- 23. Kropp P, Gerber WD. Prediction of migraine attacks using a slow cortical potential, the contingent
negative variation. *Neurosci Lett* **257**, 73-76 (1998).
- 24. Aurora SK, Ahmad BK, Welch KMA, Bhardhwaj P, Ramadan NM. Transcranial magnetic stimulation
confirms hyperexcitability of occipital cortex in migraine. *Neurology* **50**, 1111-1114 (1998).
- 25. Mulleners WM, Chronicle EP, Palmer JE, Koehler PJ, Vredeveld JW. Visual cortex excitability in
migraineurs. *Headache* **41**, 565-572 (2001).
- 26. Van der Kamp W, Maassen Van Den Brink A, Ferrari MD, Gert van Dijk J. Interictal hyperexcitability
in migraine patients demonstrated with transcranial magnetic stimulation. *Journal of the*
*Neurological Sciences* **139**, 106-110 (1996).
- 27. Young W, Shaw J, Bloom M, Gebeline-Myers C. Correlation of increase in phosphene threshold
with reduction of migraine frequency: observation of levetiracetam-treated subjects. *Headache*
**48**, 1490-1498 (2008).
- 28. Aurora SK, Barrodale P, Chronicle EP, Mulleners WM. Cortical inhibition is reduced in chronic and
episodic migraine and demonstrates a spectrum of illness. *Headache* **45**, 546-552 (2005).
- 29. Mulleners WM, Chronicle EP, Vredeveld JW, Koehler PJ. Visual cortex excitability in migraine
before and after valproate prophylaxis: a pilot study using TMS. *Eur J Neurol* **9**, 35-40 (2002).
- 30. Palermo A, Fierro B, Giglia G, Cosentino G, Puma AR, Brighina F. Modulation of visual cortex
excitability in migraine with aura: effects of valproate therapy. *Neurosci Lett* **467**, 26-29 (2009).

- 31. Ayata C, Jin H, Kudo C, Dalkara T, Moskowitz MA. Suppression of cortical spreading depression in
migraine prophylaxis. *Ann Neurol* **59**, 652-661 (2006).
- 32. Dreier JP, *et al.* Is spreading depolarization characterized by an abrupt, massive release of gibbs
free energy from the human brain cortex? *Neuroscientist* **19**, 25-42 (2013).
- 33. DeSalvo MN, *et al.* Focal BOLD fMRI changes in bicuculline-induced tonic-clonic seizures in the rat.
*Neuroimage* **50**, 902-909 (2010).
- 34. Vongerichten AN, *et al.* Characterisation and imaging of cortical impedance changes during
interictal and ictal activity in the anaesthetised rat. *Neuroimage* **124**, 813-823 (2016).
- 35. Medina-Ceja L, Ventura-Mejia C. Differential effects of trimethylamine and quinine on seizures
induced by 4-aminopyridine administration in the entorhinal cortex of vigilant rats. *Seizure* **19**,
507-513 (2010).
- 36. Schwartz TH, Bonhoeffer T. In vivo optical mapping of epileptic foci and surround inhibition in
ferret cerebral cortex. *Nat Med* **7**, 1063-1067 (2001).
- 37. Ayata C, Lauritzen M. Spreading Depression, Spreading Depolarizations, and the Cerebral
Vasculature. *Physiol Rev* **95**, 953-993 (2015).

REVIEWER COMMENTS

Reviewer #1 (Remarks to the Author):

The authors made a careful and reasonable revision according to my suggestions.

The authors did an experiment using a model of generalized epilepsy to induce seizures and SD by intraperitoneal injection of KA. It seems that when SDs were triggered by seizures, seizures were suppressed by SDs for a long time. The result confirmed the author's unifying theory to a certain extent.

The authors explained that the effect of NMDA receptor inhibitor MK-801 on epilepsy and SD may depend on the animal model and dose. I think it is reasonable.

The authors also explained the reason why FHM1 mutations do predispose to seizures and awake FHM mice also easily develop ostensibly spontaneous SDs. The explanation is reasonable.

The authors revised some mistakes in minor issues.

There are only two minor mistakes in the revised manuscript.

1. The abbreviations '4AP' and '4-AP' in Figure 3C are not uniform.

2. There is a hidden character in Figure 2B (PG). Please find it carefully.

I recommend the manuscript to publication after the two mistakes are revised.

Reviewer #2 (Remarks to the Author):

The authors were thorough and conscientious in revising the manuscript, adding new experiments and data analysis, and addressing all reviewer concerns. The manuscript presents important new concepts and data that are well described and articulated. There are no major critiques.

Minor comments:

Page 4, lines 115-117: the authors might also mention whether lower concentrations generated seizures, and if so, what proportions, durations, and magnitudes.

Page 10, line 375-377: These sentences are rather understated, as opinions portrayed by some authors, especially considering the present authors' own conclusive experiments contributing to these concepts (e.g. von Bornstadt et al., *Neuron* 85:1117-31, 2015). Should anoxic SD triggering the cytotoxic edema that leads to infarction not be considered a fact? Some references might be added here.

The co-occurrence and positive correlation of SDs and seizures in middle cerebral artery occlusion may be of interest to the authors (*JCBFM* 26(5): 696-707, 2006)

We thank the Referees again for their constructive critiques and greatly appreciate the opportunity to
respond and revise the manuscript accordingly. We took each suggestion to heart and addressed them as
outlined point-by-point below. All revisions are marked by red font in the text.

**Reviewer #1:**

*1.The abbreviations '4AP' and '4-AP' in Figure 3C are not uniform.*

Done, thank you.

*2.There is a hidden character in Figure 2B (PG). Please find it carefully.*

Done, thank you.

**Reviewer #2:**

*Minor comments:*

*Page 4, lines 115-117: the authors might also mention whether lower concentrations generated seizures,*
*and if so, what proportions, durations, and magnitudes.*

We revised the sentence as follows: “Lower concentrations of 4AP (5 or 30 mM) or BIC (0.005, 0.5 and 1
mM) **led to smaller and shorter lasting concentration dependent ECoG power density elevations** but did
not trigger SD (n=9 total; data not shown).” We opted not to present quantitative data because of smaller
sample sizes in these experiments.

*Page 10, line 375-377: These sentences are rather understated, as opinions portrayed by some authors,*
*especially considering the present authors' own conclusive experiments contributing to these concepts*
*(e.g. von Bornstadt et al., Neuron 85:1117-31, 2015). Should anoxic SD triggering the cytotoxic edema that*
*leads to infarction not be considered a fact? Some references might be added here.*

We agree, which is why we had the following later in the same paragraph: “There are many physiological
(i.e., part of normal function, or beneficial) processes that can be pathological when they are overactive
or occur at the wrong place and time, and SD might be one of those. When SD occurs in the wrong context
(e.g., brain injury), it can have pathological consequences.”

*The co-occurrence and positive correlation of SDs and seizures in middle cerebral artery occlusion may be*
*of interest to the authors (JCBFM 26(5): 696-707, 2006).*

We thank the Reviewer for this citation, which we have read numerous times in the past and consider as
one of two critical pieces of work in our understanding of delayed onset peri-infarct SDs after ischemic
stroke in unanesthetized animals. As we have stated in our response to previous critiques, however, we
have tried to minimize our references and inferences to and from previous work on *injured brain*, because
our model did not involve brain injury where there is a complex interplay of numerous processes such as
hemodynamics. Nevertheless, we agree that the manuscript portrays an important parallel in that
occurrence of SD or SD clusters coincides with termination and relative suppression of seizure activity,
respectively. Therefore, we now cite the manuscript in Discussion (page 10, last paragraph).